# Differentially Private Learning with Margin Guarantees

**Raef Bassily**
The Ohio State University
& Google Research NY
bassily.1@osu.edu

**Mehryar Mohri**
Google Research
& Courant Institute
mohri@google.com

**Ananda Theertha Suresh**
Google Research, NY
theertha@google.com

## Abstract

We present a series of new differentially private (DP) algorithms with dimension-independent margin guarantees. For the family of linear hypotheses, we give a pure DP learning algorithm that benefits from relative deviation margin guarantees, as well as an efficient DP learning algorithm with margin guarantees. We also present a new efficient DP learning algorithm with margin guarantees for kernel-based hypotheses with shift-invariant kernels, such as Gaussian kernels, and point out how our results can be extended to other kernels using oblivious sketching techniques. We further give a pure DP learning algorithm for a family of feed-forward neural networks for which we prove margin guarantees that are independent of the input dimension. Additionally, we describe a general label DP learning algorithm, which benefits from relative deviation margin bounds and is applicable to a broad family of hypothesis sets, including that of neural networks. Finally, we show how our DP learning algorithms can be augmented in a general way to include model selection, to select the best confidence margin parameter.

## 1 Introduction

Preserving privacy is a crucial objective for machine learning algorithms. A widely adopted criterion in statistical data privacy is the notion of differential privacy (DP) [DMNS06, Dwo06, DR14], which ensures that the information gained by an adversary is roughly invariant to the presence or absence of an individual in a dataset. Despite the remarkable theoretical and algorithmic progress in differential privacy over the last decade or more, however, its application to learning still faces several obstacles. A recent series of publications have shown that differentially private PAC learning of infinite hypothesis sets is not possible, even for common hypothesis sets such as that of linear functions. In fact, this is the case for any hypothesis set containing threshold functions [BNSV15, ALMM19]. These results imply serious limitations for private agnostic learnability.

Another rich body of literature has studied differentially private empirical risk minimization (DP-ERM) and differentially private stochastic convex optimization (DP-SCO) (e.g., [CMS11, JT14, BST14, BFTT19, FKT20, SSTT21, BGN21, AFKT21, BGM21]). When the underlying optimization problem is constrained (*constrained setting*), tight upper and lower bounds have been derived for the excess empirical risk of DP-ERM [BST14] and for the excess population risk for DP-SCO [BFTT19, FKT20]. These results show that learning guarantees necessarily admit a dependency on the dimension $d$ of the form $\sqrt{d}/m$, where $m$ is the sample size. This dependency is persistent, even in the special case of *generalized linear losses* (GLLs) [BST14], which limits the benefit of such guarantees, since learning algorithms typically deal with high-dimensional spaces.

When the underlying optimization problem is unconstrained (*unconstrained setting*) and the loss is a generalized linear loss, the bounds given by [JT14], [SSTT21] and [BGM21] are dimension-independent but they admit a dependency on $\|w^*\|^2$, where $w^*$ is the unconstrained minimizer of the

36th Conference on Neural Information Processing Systems (NeurIPS 2022).

expected loss (population risk), or $\|\widehat{w}\|^2$, where $\widehat{w}$ is the unconstrained minimizer of the empirical loss. Since the problem is unconstrained, the norm of these vectors can be very large, even for classification problems for which the minimizer of the zero-one loss admits a relatively small norm. Thus, in both the constrained and unconstrained settings, the learning guarantees derived from DP-ERM and DP-SCO are weak for hypothesis sets commonly used in machine learning.

The results just mentioned raise some fundamental questions about private learning: is differentially private learning with favorable (dimension-independent) guarantees possible for standard hypothesis sets? Must one resort to distribution-dependent bounds instead? In view of the negative PAC-learning results and other learning bounds mentioned earlier, we will seek instead optimistic margin-based learning bounds.

In the context of classification, learning bounds for linear hypotheses based on the dimension or, more generally, based on the VC-dimension of the hypothesis set are known to be too pessimistic since they deal with the worst case. Instead, margin bounds have been shown to be the most informative and useful guarantees [KP02, SFBL97]. This motivates our study of differentially private learning algorithms with margin-based guarantees. Note that our *confidence-margin* analysis and guarantees do not require the hard-margin separability assumptions adopted in [BDMN05, LNUZ20], which is a strong assumption that typically does not hold in practice. Another existing study that deals with somewhat related questions is that of [CHS14]. But, the paper deals with a specific class of maximization problems and adopts a non-standard definition of margin. Another related line of work is that of [RBHT09] and [CMS11] on DP Kernel classifiers. These works either provide suboptimal, dimension-dependent learning guarantees or make strong assumptions about the Fourier coefficients of the kernel predictors. We discuss these prior works in more detail in Section 1.1.

**Main contributions.** We present a series of new differentially private (DP) algorithms for learning linear classifiers, kernel classifiers, and neural-network classifiers with dimension-independent, confidence-margin guarantees. In Section 3, we study the family of linear hypotheses. We first give a pure DP learning algorithm with relative deviation margin guarantees. Next, we present an efficient DP learning algorithm with margin guarantees. Our algorithm is based on a faster construction for the JL-transform and a faster DP-ERM algorithm. While the general structure of our algorithms for linear classifiers is similar to that of [LNUZ20], our results require a new analysis that takes into account the scale-sensitive nature of the margin loss and the $\rho$-hinge loss. In Section 4, We present a new efficient DP learning algorithm with margin guarantees for kernel-based hypothesis sets, assuming that the positive definite kernel used is shift-invariant, as with the most commonly used Gaussian kernels. Our algorithm combines kernel approximation with the use of the JL-transform. Our result is based on a new style of analysis that uses regularized ERM as a reference. These ideas enable us to attain a bound that nearly matches the non-private margin bound, without resorting to the strong assumptions in prior work. Our confidence-margin bounds for DP learning of linear and kernel classifiers nearly match the standard, non-private confidence-margin bounds. In Section 5, we initiate the study of DP learning of neural networks with margin guarantees. We design a pure DP learning algorithm for a family of feed-forward neural networks for which we prove a confidence-margin bound that is independent of the input dimension and exhibits better dependence on the network parameters than the bounds attained via uniform convergence. Our result entails a new analysis of embedding-based "network compression" technique. Our margin bound for neural networks is the first of its type. The bound is independent of the input dimension and scales only linearly with the number of activation units. In Appendix E, We further present a *label privacy* learning algorithm, which we show benefits from relative deviation margin bounds. The algorithm and its guarantee are applicable to a broad family of hypothesis sets, including that of neural networks. Finally, we show in Appendix F how our DP learning algorithms can be augmented in a general way to include model selection, to select the best confidence margin parameter.

## 1.1 Related work

**Prior work on unconstrained GLLs.** [JT14] and [SSTT21] showed that it is possible to derive dimension-independent risk bounds for DP-ERM and DP-SCO in the context of linear prediction, when the parameter space is unconstrained and the loss function is convex and Lipschitz (GLL). However, their bounds scale with $\|w^*\|$, the norm of the optimal unconstrained minimizer of the expected surrogate loss such as the hinge loss. Also, using their techniques for unconstrained DP-ERM for GLLs together with uniform convergence would yield generalization error bounds that scale with the norm of the unconstrained empirical risk minimizer $\widehat{w}$. The first issue with this line of work is that the norms of such unconstrained solutions can be very large, thereby resulting in uninformative

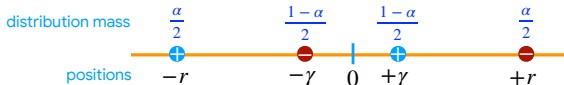

**Figure 1:** Simple example in dimension one for which the minimizer of the expected hinge loss $\mathbb{E}[\ell^{\mathsf{hinge}}(w)]$ is $w^* = \frac{1}{\gamma}$ and thus $\|w^*\| = \frac{1}{\gamma} \gg 1$ for $\gamma \ll 1$. Here, any other $w > 0$, in particular with a small norm, achieves the same zero-one loss as $w^*$.

bounds. In fact, one can construct simple, low-dimensional examples, where $\|w^*\| = \Omega(m)$ while there is a predictor $w$ with $\|w\| = O(1)$ that attains the same expected zero-one error, see Figure 1 (a detailed analysis of that example is given in Appendix H). More importantly, the paradigm adopted in this line of work is to first devise an algorithm and next derive bounds for its excess risk. In contrast, we start from strong generalization error bounds, which we use to guide the design of our algorithm.

**Prior work on DP learning of hard-margin halfspaces.** [BDMN05] and [LNUZ20] studied DP learning of linear classifiers in the separable setting, that is with a hard- or *geometric margin*. [BDMN05] gave a construction based on a private version of the Perceptron algorithm, which results in a dimension-dependent bound on the expected error. This result was later improved by [LNUZ20] who gave new constructions with dimension-independent guarantees based their nice idea of using embeddings, namely, the Johnson-Lindenstrauss (JL) transform, to reduce the dimensionality of the problem from $d$ to $1/\gamma$, where $\gamma$ is the geometric margin. Note that the hard-margin separability is a strong assumption that typically does not hold in practice. Moreover, the constructions proposed in [LNUZ20] require the knowledge of the margin for their guarantees to be valid. In contrast, our work considers the more general notion of *confidence margin*, which does not require the existence of a geometric margin and applies to realistic scenarios with non-separable data. Moreover, the confidence-margin parameter, $\rho$, in our algorithms is tunable and can be optimized. Importantly, our algorithms still yield meaningful learning guarantees even if this parameter is not optimized. Our algorithms for linear classifiers also make use of an embedding as a pre-processing step. However despite the similar structure to that of [LNUZ20], our algorithm requires a new analysis and different settings of parameters. This new analysis is necessary to deal with the scale-sensitive nature of our bounds, due to the absence of a hard-margin.

**Prior work on DP Kernel classifiers.** [RBHT09] were the first to provide differentially private constructions for SVMs in both the finite-dimensional feature space and kernel settings. However, their constructions are suboptimal and the resulting bounds suffer from a polynomial dependence on the dimension of the feature space. In addition, their error bound has a sub-optimal dependence on the sample size $m$ and also has an explicit dependence on the $\ell_1$-norm of the dual variables of the SVM. In general, this norm can be as large as $\sqrt{m}$, in which case their error bound becomes vacuous. [CMS11] gave a similar construction for shift-invariant kernels. However, their error guarantees are based on the kernel approximation results of [RR08] and hence entail a relatively strong condition on the Fourier coefficients of the kernel predictors and the kernel density. We note that the standard assumption of bounded Reproducing Kernel Hilbert Space (RKHS) norm does not imply such a condition. [JT13] gave algorithms for DP predictions with kernels, where the goal is to privately generate predictions (labels) on a small test set that admits no privacy constraints. In such scenarios, their algorithms do not output a classifier. [JT13] also gave a construction for private learner that outputs a kernel classifier, however, their construction is computationally inefficient and the resulting error guarantees are dimension-dependent.

## 2 Preliminaries

We consider an input space $\mathcal{X}$, a binary output space $\mathcal{Y} = \{-1, +1\}$ and a hypothesis set $\mathcal{H}$ of functions mapping from $\mathcal{X}$ to $\mathbb{R}$. We denote by $\mathcal{D}$ a distribution over $\mathcal{Z} = \mathcal{X} \times \mathcal{Y}$ and denote by $R_{\mathcal{D}}(h)$ the generalization error and by $\widehat{R}_S(h)$ the empirical error of a hypothesis $h \in \mathcal{H}$:

$$R_{\mathcal{D}}(h) = \mathop{\mathbb{E}}_{z=(x,y)\sim\mathcal{D}}[1_{yh(x)\leq0}] \qquad \widehat{R}_S(h) = \mathop{\mathbb{E}}_{z=(x,y)\sim S}[1_{yh(x)\leq0}],$$

where we write $z \sim S$ to indicate that $z$ is randomly drawn from the empirical distribution defined by $S$. Given $\rho \geq 0$, we similarly define the $\rho$-margin loss and empirical $\rho$-margin loss of $h \in \mathcal{H}$:

$$R_{\mathcal{D}}^{\rho}(h) = \mathop{\mathbb{E}}_{z=(x,y)\sim\mathcal{D}}[1_{yh(x)\leq\rho}] \qquad \widehat{R}_S^{\rho}(h) = \mathop{\mathbb{E}}_{z=(x,y)\sim S}[1_{yh(x)\leq\rho}].$$

The $\rho$-margin loss is not convex. Hence, we also consider the $\rho$-hinge loss to provide computationally-efficient algorithms. For any $\rho > 0$, define the $\rho$-hinge loss as $\ell^\rho(u) \triangleq \max(1 - u/\rho, 0)$, $u \in \mathbb{R}$. Similar to the above definitions, given $\rho > 0$, for a sample $S$, we define the expected and the empirical $\rho$-hinge losses as follows:

$$L_\mathcal{D}^\rho(w) = \mathop{\mathbb{E}}_{z=(x,y)\sim\mathcal{D}}[\ell^\rho(y_i\langle w, x_i\rangle)] \qquad \widehat{L}_S^\rho(w) = \mathop{\mathbb{E}}_{z=(x,y)\sim S}[\ell^\rho(y_i\langle w, x_i\rangle)].$$

In the context of learning, differential privacy is defined as follows.

**Definition 2.1** (Differential privacy). *Let $\varepsilon, \delta \geq 0$. Let $\mathcal{A}: (\mathcal{X} \times \mathcal{Y})^m \to \mathcal{H}$ be a randomized algorithm. We say that $\mathcal{A}$ is $(\varepsilon, \delta)$-DP if for any measurable subset $O \subset \mathcal{H}$ and all $S, S' \in (\mathcal{X} \times \mathcal{Y})^m$ that differ in one sample, the following inequality holds:*

$$\mathbb{P}(\mathcal{A}(S) \in O) \leq e^\varepsilon \mathbb{P}(\mathcal{A}(S') \in O) + \delta. \tag{1}$$

*If $\delta = 0$, we refer to this guarantee as* pure differential privacy.

# 3 Private Algorithms for Linear Classification with Margin Guarantees

In this section, we present two private learning algorithms for linear classification with margin guarantees: first, a computationally inefficient pure DP algorithm, which we show benefits from relative deviation margin bounds, next, a computationally efficient DP algorithm with a dimension-independent bound expressed in terms of the empirical $\rho$-hinge loss.

Let $\mathbb{B}^d(r) \triangleq \{x \in \mathbb{R}^d : \|x\|_2 \leq r\}$ denote the Euclidean ball in $\mathbb{R}^d$ of radius $r$ and let $\mathcal{X} \subseteq \mathbb{B}^d(r)$ denote the feature space. We will use the shorthand $\mathbb{B}^d$ for $\mathbb{B}^d(1)$. We consider the class of linear predictors over $\mathcal{X}$ defined by $\mathcal{H}_{\mathsf{Lin}} = \{h_w : x \mapsto \langle w, x\rangle \mid w \in \mathbb{B}^d(\Lambda)\}$. Note that one can represent the general class of affine functions over $\mathbb{R}^d$ as linear functions over $\mathbb{R}^{d+1}$ by simply mapping each $x \in \mathbb{R}^d$ to $\tilde{x} = (x, 1) \in \mathbb{R}^{d+1}$. Thus, without loss of generality, we will consider $\mathcal{H}_{\mathsf{Lin}}$. Here, we view $d$ as possibly much larger than the sample size $m$. We also note that even though the predictors in the input class $\mathcal{H}_{\mathsf{Lin}}$ admit $\Lambda$-bounded norm, we do not constrain the algorithm to output a predictor with bounded norm, which circumvents the necessary dependence on the dimension in constrained DP optimization [BST14].

## 3.1 Pure DP Algorithm for Linear Classification

A standard method for designing differentially private algorithms for a continuous hypothesis class is to apply the exponential mechanism [MT07] to a cover of the hypothesis class. Since $\mathcal{H}_{\mathsf{Lin}}$ is $d$-dimensional, the size of a useful cover is about $2^{\Omega(d)}$, thus, a direct application of the exponential mechanism yields an $\Omega(d)$-bound; we give a simple example illustrating that in Appendix G. Thus, instead, we seek to reduce the size of the cover without impacting its accuracy, using random projections. This results in a mapping $\Phi$ from $\mathbb{R}^d$ to a lower-dimensional space $\mathbb{R}^k$.

For linear classification, we wish to preserve intra-point distances and angles, that is $x \cdot x' \approx \Phi x \cdot \Phi x'$ for points $x$ and $x'$. It is known that this property can be fulfilled as a corollary of the Johnson-Lindenstrauss lemma [Nel11, Theorem 109]. For completeness, we provide a full proof in Appendix A. More interestingly, we show that we can reduce the dimension to $\widetilde{O}(\Lambda^2 r^2/\rho^2)$, without the error decreasing significantly. We then run the exponential mechanism in this lower-dimensional space and next compute a classifier $\tilde{w}$ in that space. We finally derive a classifier in the original space by applying the transpose of the original projection matrix $\Phi^T\tilde{w}$. Note that the final output $\Phi^T\tilde{w}$ has expected norm $\widetilde{O}\left(\frac{\sqrt{d}\rho}{r}\right)$ and may not lie in $\mathbb{B}^d(\Lambda)$.

Algorithm 1 gives the pseudocode of the full algorithm. The algorithm and the analysis in this section include a dimensionality reduction technique for mapping the feature vectors from the input $d$-dimensional space to a $k$-dimensional space, where $k = \widetilde{O}(\Lambda^2 r^2/\rho^2)$ for some $\rho \in (0, \Lambda r]$. Hence, we will be dealing with "compressed" parameter vectors in $\mathbb{R}^k$. To distinguish these two spaces, we will denote the empirical error and the empirical $\rho$-margin error in this $k$-dimensional space as $\widehat{R}_{S'}^{(k)}(w')$ and $\widehat{R}_{S'}^{(k),\rho}(w')$, respectively, where $w' \in \mathbb{B}^k$ and $S' \in (\mathbb{R}^k \times \mathcal{Y})^m$.

**Algorithm 1** $\mathcal{A}_{\mathsf{PrivMrg}}$: Private Learner of Linear Classifiers with Margin Guarantees

---

**Require:** Dataset $S = ((x_1, y_1), \ldots, (x_m, y_m)) \in (\mathcal{X} \times \{\pm 1\})^m$; privacy parameter $\varepsilon > 0$; margin parameter $\rho \in (0, \Lambda r]$; confidence parameter $\beta > 0$.

1: Let $k = O\left(\dfrac{\Lambda^2 r^2 \log\left(\frac{m}{\beta}\right)}{\rho^2}\right)$.

2: Let $\Phi$ be a random $k \times d$ matrix with entries drawn i.i.d. uniformly from $\{\pm\frac{1}{\sqrt{k}}\}$.

3: Let $S_\Phi = \{(x_\Phi, y) : (x, y) \in S\}$, where for any $x \in \mathbb{R}^d$, $x_\Phi \triangleq \Phi x \in \mathbb{R}^k$.

4: Let $\mathcal{C}$ be a $\frac{\rho}{10 r}$-cover of $\mathbb{B}^k(2\Lambda)$.

5: Run the Exponential mechanism over $\mathcal{C}$ with privacy parameter $\varepsilon$, sensitivity $1/m$, and score function $-\widehat{R}^{(k)}_{S_\Phi}(w)$ for $w \in \mathcal{C}$, to select $\tilde{w} \in \mathcal{C}$.

6: **return** $w^{\mathsf{Priv}} = \Phi^\top \tilde{w}$, where $\Phi^\top$ denotes the transposition of $\Phi$.

---

**Theorem 3.1.** *Algorithm 1 is $\varepsilon$-differentially private. For any $\beta \in (0, 1)$, with probability at least $1 - \beta$ over the draw of a sample $S$ of size $m$ from $\mathcal{D}$, the solution $w^{\mathsf{Priv}}$ it returns satisfies:*

$$R_\mathcal{D}(w^{\mathsf{Priv}}) \leq \min_{w \in \mathbb{B}^d(\Lambda)} \left\{ \widehat{R}^\rho_S(w) + \widetilde{O}\left( \sqrt{\widehat{R}^\rho_S(w) \frac{\Lambda^2 r^2}{m\rho^2}} + \frac{\Lambda^2 r^2}{\rho^2 \min(1, \varepsilon) m} \right) \right\}.$$

The proof is given in Appendix B.1. The theorem provides a *margin guarantee* for the private solution. Note that no assumption is made about separability or the existence of a favorable hard-margin. Instead, through the first term, the bound is based on the distribution of the empirical margins $y(w \cdot x)$. The theorem suggests that, when the empirical $\rho$-margin loss remains small for a relatively large value of the confidence margin parameter $\rho$, then $w^{\mathsf{Priv}}$ benefits from a strong generalization guarantee. These comments hold similarly for other margin bounds presented in this paper.

This result, although given for a computationally inefficient method, is stronger than several previously known ones: First, it is an $(\varepsilon, 0)$-pure differential privacy guarantee; second, it is dimension-independent and furthermore, unlike prior work, the norm of the optimal hypothesis does not appear in the bound. Furthermore, since it is a relative deviation margin bound, it smoothly interpolates between the realizable case of $\widehat{R}^\rho_S(w) = 0$ and the case of $\widehat{R}^\rho_S(w) > 0$. For a sample of size $m$, the bound is based on an interpolation between a $1/\sqrt{m}$-term that includes the square-root of the empirical margin loss as a factor and another term in $1/m$. In particular, when the empirical margin loss is zero, the bound only admits the $1/m$ fast rate term. As a corollary, note that, up to constants, one can always obtain privacy for $\varepsilon > 1$ essentially for free.

**Dependence on $\Lambda/\rho$:** Our bound depends on the choice of $\Lambda/\rho$. We note that this is fundamentally different from the dependence on $\|w^*\|$ in prior works on GLLs [JT14, SSTT21] for two reasons: First, unlike $\|w^*\|$, $\Lambda/\rho$ is a measurable and, more importantly, tunable quantity, which we can select an optimal setting for[1] (as we do in Appendix F). Second, the optimal choice for this parameter can be much smaller than $\|w^*\|$ as demonstrated in Appendix H.

## 3.2 Efficient Private Algorithm for Linear Classification

The $\rho$-margin loss is not convex and it is known that its minimization is generally intractable. Instead, we devise a computationally efficient algorithm, whose guarantees are expressed in terms of the empirical $\rho$-hinge loss. Algorithm 2 shows the pseudocode of our algorithm. We now discuss the key steps of the algorithm.

**Fast JL-transform.** Our algorithm entails a dimensionality reduction step (step 3) as in Algorithm 1. Here, we note that the new dimension $k$ is chosen to be $\widetilde{O}(m\varepsilon)$, which enables us to control the influence of the dimensionality reduction on the empirical hinge loss. We also note that this step is carried out via a fast construction for the JL-transform (Lemma A.4), which takes $O(d \log(d))$ time, assuming $d > \varepsilon m$.

---

[1] Note that, without loss of generality, we can set $\Lambda = 1$ and optimize only with respect to $\rho$.

**Algorithm 2** $\mathcal{A}_{\mathsf{EffPrivMrg}}$: Efficient Private Learner of Linear Classifiers with Margin Guarantees

---

**Require:** Dataset $S = ((x_1, y_1), \dots, (x_m, y_m)) \in (\mathbb{B}^d(R) \times \{\pm 1\})^m$; privacy parameters $\varepsilon, \delta$; norm bound $\Lambda$; margin parameter $\rho \in (0, \Lambda r]$; confidence parameter $\beta > 0$.
1: Let $k = \frac{\varepsilon m \log(m/\beta)}{\log^{\frac{3}{2}}(1/\delta) \log(1/\beta)}$.
2: Let $\Phi$ be a random $k \times d$ matrix from the construction in Lemma A.4.
3: Let $S_\Phi = \{(\Pi_{\mathbb{B}^k(2r)}(x_\Phi), y) : (x, y) \in S\}$, where for any $x \in \mathbb{R}^d$, $x_\Phi \triangleq \Phi x \in \mathbb{R}^k$ and $\Pi_{\mathbb{B}^k(2r)}$ is the Euclidean projection on $\mathbb{B}^k(2r)$.
4: Privately solve the convex ERM problem: $\underset{w \in \mathbb{B}^k(2\Lambda)}{\operatorname{argmin}} \widehat{L}^\rho_{S_\Phi}(w)$ via Algorithm 4 (Appendix B.2) and return $\tilde{w} \in \mathbb{B}^k(2\Lambda)$.
5: **return** $w^{\mathsf{Priv}} = \Phi^\top \tilde{w}$, where $\Phi^\top$ denotes the transposition of $\Phi$.

---

**Near linear-time DP convex ERM.** After this step, we invoke an efficient algorithm for DP-ERM (step 4 in Algorithm 2) to find an approximate minimizer of the empirical $\rho$-hinge loss $\widehat{L}^\rho_{S_\Phi}(w)$, rather than using the exponential mechanism to find an approximate minimizer for the empirical zero-one loss $\widehat{R}_{S_\Phi}(w)$. To improve the running time of step 4, we use the construction in [BGM21, Algorithm 2] to solve DP-ERM in near-linear time and with high-probability guarantee on the excess empirical risk (see Algorithm 4 in Appendix B.2). The algorithm of [BGM21] is devised for DP-SCO with respect to non-smooth generalized linear losses. It is based on a combination of a smoothing technique via proximal steps and the phased SGD algorithm [FKT20, Algorithm 2] for smooth DP-SCO. The algorithm of [BGM21] can be used for DP-ERM if it is fed with a sample from the empirical distribution of the dataset. However, the privacy guarantee requires a careful privacy analysis to deal with the fact that this sample may contain duplicate entries from the original dataset.

Moreover, since the original algorithms in [FKT20, BGM21] provide only expectation guarantees and we aim at high-probability learning bounds, we use a standard private confidence-boosting technique to provide a high-probability guarantee on the excess risk of our variant. We summarize the guarantees of this variant in the following lemma. The details of the construction and the proof of the lemma below can be found in Appendix B.2.

**Lemma 3.1.** *Let* $m \in \mathbb{N}$, $0 < \delta < \frac{1}{m}$, *and* $0 < \varepsilon \leq \log(1/\delta)$. *Algorithm 4 (Appendix B.2) is* $(\varepsilon, \delta)$-*DP. Let* $\beta \in (0, 1)$. *Let* $k \in \mathbb{N}$, *and* $\tilde{r}, \Lambda > 0$. *Let* $\widetilde{S} \in (\mathbb{B}^k(\tilde{r}) \times \{\pm 1\})^m$ *be the input dataset and* $\mathbb{B}^k(\Lambda)$ *be the parameter space. With probability* $1 - \beta$ *over the randomness in Algorithm 4, the output* $\tilde{w}$ *satisfies*

$$\widehat{L}^\rho_{\widetilde{S}}(\tilde{w}) \leq \min_{w \in \mathbb{B}^k(\Lambda)} \widehat{L}^\rho_{\widetilde{S}}(w) + \frac{\Lambda \tilde{r}}{\rho} \cdot O\left(\frac{1}{\sqrt{m}} + \frac{\sqrt{k} \log^{\frac{3}{2}}(\frac{1}{\delta}) \log(\frac{1}{\beta})}{\varepsilon m}\right).$$

*Moreover, Algorithm 2 requires* $O(m \log(m) \log(1/\beta))$ *gradient computations.*

We now state our main result in this section, which we prove in Appendix B.3.

**Theorem 3.2.** *Let* $0 < \delta < \frac{1}{m}$ *and* $0 < \varepsilon \leq \log(1/\delta)$. *Algorithm 2 is* $(\varepsilon, \delta)$-*DP. Let* $\beta \in (0, 1)$. *Let* $S \sim \mathcal{D}^m$ *for a distribution* $\mathcal{D}$ *over* $\mathbb{B}^d(r) \times \{\pm 1\}$. *Algorithm 2 outputs* $w^{\mathsf{Priv}} \in \mathbb{R}^d$ *such that with probability at least* $1 - \beta$, *we have*

$$R_{\mathcal{D}}(w^{\mathsf{Priv}}) \leq \min_{w \in \mathbb{B}^d(\Lambda)} \widehat{L}^\rho_S(w) + \widetilde{O}\left(\frac{\Lambda r}{\rho \sqrt{\min(1, \varepsilon) \, m}}\right).$$

*Moreover, Algorithm 2 runs in time* $O\left(md \log(\max(d, m)) + \varepsilon m^2 \log(m)/\log^{\frac{3}{2}}(1/\delta)\right)$.

## 4 Private Algorithms for Kernel-Based Classifiers with Margin Guarantees

In this section, we present private algorithms with margin guarantees for kernel-based predictors [SS02, STC04]. We first consider a continuous, positive definite, shift-invariant kernel $K : \mathcal{X} \times \mathcal{X} \to \mathbb{R}$, where $K(x, x) = r^2$ for all $x \in \mathcal{X}$. The associated feature map is defined as $\psi(x) \triangleq K(\cdot, x)$, $x \in \mathcal{X}$, where $\mathcal{X} \subseteq \mathbb{B}^d(r)$.

**Overview of the technique.** Our approach is based on approximating the feature map $\psi$ by a finite-dimensional feature map $\widehat{\psi} : \mathcal{X} \to \mathbb{B}^{2D}(r)$ determined via Random Fourier Features (RFFs). The dimension $2D$ of the approximate feature map is chosen to be sufficiently large to ensure that for all pairs of feature vectors $x_i, x_j$ in a training set $S = ((x_1, y_1), \ldots, (x_m, y_m))$, we have $|\langle \widehat{\psi}(x_i), \widehat{\psi}(x_j) \rangle - K(x_i, x_j)| \lesssim \frac{1}{m}$ with high probability over the randomness of $\widehat{\psi}$ (due to RFFs). This suffices to derive an upper bound (margin bound) on the true error of a finite-dimensional linear predictor trained on the sample made of the labeled points $(\widehat{\psi}(x), y)$, $(x, y) \in S$, that is essentially the same as the margin bound known for the kernel classifier. Hence, in effect, we reduce the problem to that of learning a linear classifier in a $2D$-dimensional space, which we can solve privately using Algorithm 2. Note that the output predictor in this case is a finite-dimensional linear function rather than a function in the Reproducing Kernel Hilbert Space. A full description of our DP learner of kernel classifiers is given in Algorithm 3 below.

**Bochner's Theorem and RFFs.** Since the kernel $K$ is shift-invariant, it can be expressed as $K(x, x') = r^2 \bar{K}(x - x'), x, x' \in \mathcal{X}$ for some function $\bar{K} : \mathcal{Z} \to \mathbb{R}$, where $\mathcal{Z} = \{z = x - x' : x, x' \in \mathcal{X}\}$. Moreover, since $K$ is positive-definite, $\bar{K}$ is the Fourier transform of a probability distribution $P_{\bar{K}}$:

$$\bar{K}(x) = \int_{\mathcal{X}} e^{i \langle \omega, x \rangle} P_{\bar{K}}(\omega) d\omega.$$

This follows from Bochner's Theorem [Rud17]. Random Fourier Features (RFFs) provide a simple method introduced in [RR07] to approximate kernel feature maps in a data-independent fashion. The idea is based on Bochner's theorem. In particular, we first sample $\omega_1, \ldots, \omega_D$ independently from the probability distribution $P_{\bar{K}}$. Then, we define an approximate feature map as follows:

$$\widehat{\psi}(x) \triangleq \frac{r}{\sqrt{D}} \left( \cos \langle \omega_1, x \rangle, \sin \langle \omega_1, x \rangle, \ldots \cos \langle \omega_D, x \rangle, \sin \langle \omega_D, x \rangle \right), \quad \forall x \in \mathcal{X}. \tag{2}$$

For $D$ sufficiently large, it can be shown that $\langle \widehat{\psi}(x), \widehat{\psi}(x') \rangle$ concentrates around $K(x, x')$ for all pairs $x, x' \in \mathcal{X}$ [MRT18, Theorem 6.28]. In our analysis below, we only need that concentration to hold uniformly over pairs $x, x'$ from a fixed training set rather than uniformly over all pairs $x, x' \in \mathcal{X}$. This leads to a simpler approximation guarantee, which we formally state below.

**Theorem 4.1.** *Let $S_{\mathcal{X}} = (x_1, \ldots, x_m) \in \mathcal{X}^m$. Let $K$ be a shift-invariant, positive definite kernel, where $K(x, x) = r^2$, $\forall x \in \mathcal{X}$. Let $P_{\bar{K}}$ be the probability distribution associated with $K$. Suppose $\omega_1, \ldots, \omega_D$ are drawn independently from $P_{\bar{K}}$. With probability 1, we have $\|\widehat{\psi}(x)\|_2 = r$, $\forall x \in \mathcal{X}$. For any $\beta \in (0, 1)$, with probability at least $1 - \beta$, for all $i, j \in [m]$ such that $i \neq j$ we have*

$$\left| \langle \widehat{\psi}(x_i), \widehat{\psi}(x_j) \rangle - K(x_i, x_j) \right| \leq 2r^2 \sqrt{\frac{\log\left(\frac{m}{\beta}\right)}{D}}.$$

*Proof.* The first assertion related to $\|\widehat{\psi}(x)\|_2 \ \forall x \in \mathcal{X}$ follows directly from the definition of $\widehat{\psi}(x)$ and a basic trigonometric identity. The proof of the second assertion about the inner products follows from the identity $\mathbb{E}_{\omega_1, \ldots, \omega_D} \left[ \langle \widehat{\psi}(x_i), \widehat{\psi}(x_j) \rangle \right] = K(x_i, x_j)$ that holds for all $x_i, x_j$, the application of Hoeffding's bound combined with the union bound over all $\approx m^2$ pairs $x_i, x_j \in S_{\mathcal{X}}$. The unbiasedness of $\langle \widehat{\psi}(x_i), \widehat{\psi}(x_j) \rangle$ follows from the fact that the expectation is the Fourier transform of $r^2 P_{\bar{K}}(\omega)$, which, by Bochner's Theorem, is $r^2 \bar{K}(x_i - x_j) = K(x_i, x_j)$. $\qquad \square$

We now state our main result, which we prove in Appendix C.

**Theorem 4.2.** *Let $r > 0$. Let $K : \mathcal{X} \times \mathcal{X} \to \mathbb{R}$ be a shift-invariant, positive definite kernel, where $K(x, x) = r^2$ for all $x \in \mathcal{X}$. For any $\varepsilon > 0$ and $\delta \in (0, 1)$, Algorithm 3 is $(\varepsilon, \delta)$-differentially private. Define $\mathcal{H}_\Lambda \triangleq \{h \in \mathbb{H} : \|w\|_{\mathbb{H}} \leq \Lambda\}$, where $\|\cdot\|_{\mathbb{H}}$ is the norm corresponding to the reproducing kernel Hilbert space (RKHS) $\mathbb{H}$ associated with the kernel $K$. Let $\beta \in (0, 1)$. Given an input sample $S$ of $m$ examples drawn i.i.d. from a distribution $\mathcal{D}$ over $\mathcal{X} \times \{\pm 1\}$, Algorithm 3 outputs $h_w^{\widehat{\psi}}$Priv such that with probability at least $1 - \beta$, we have*

$$R_{\mathcal{D}}(h_{w^{\mathrm{Priv}}}^{\widehat{\psi}}) \leq \min_{h \in \mathcal{H}_\Lambda} \widehat{L}_S^\rho(h) + \widetilde{O}\left( \frac{\Lambda r}{\rho \sqrt{\min(1, \varepsilon) \, m}} \right),$$

---

**Algorithm 3** $\mathcal{A}_{\mathsf{PrivKerMrg}}$: Efficient Private Learner of Kernel Classifiers with Margin Guarantees

---

**Require:** Dataset $S = ((x_1, y_1), \ldots, (x_m, y_m)) \in (\mathcal{X} \times \{\pm 1\})^m$; shift-invariant, positive definite kernel $K : \mathcal{X} \times \mathcal{X} \to \mathbb{R}$ with $K(x, x) = r^2$ for all $x \in \mathcal{X}$, privacy parameters $\varepsilon, \delta$; Reproducing kernel Hilbert space (RKHS) norm bound $\Lambda$; margin parameter $\rho \in (0, 2\Lambda r]$; confidence parameter $\beta > 0$.

 1: Let $P_{\bar{K}}$ be the probability distribution associated with $K$.
 2: Let $D = m^2 \log(2m/\beta)$.
 3: Draw $\omega_1, \ldots, \omega_D$ independently from $P_{\bar{K}}$.
 4: Let $S_{\widehat{\psi}} = \big( (\widehat{\psi}(x_i), y_i) : i \in [m] \big)$, where $\widehat{\psi}$ is as defined in (2).
 5: $w^{\mathsf{Priv}} \leftarrow \mathcal{A}_{\mathsf{EffPrivMrg}} \big( S_{\widehat{\psi}}, \varepsilon, \delta, 2\Lambda, \rho, \beta/2 \big)$, where $\mathcal{A}_{\mathsf{EffPrivMrg}}$ is the private learner described in Algorithm 2. Note that the input dimension to $\mathcal{A}_{\mathsf{EffPrivMrg}}$ is $2D$ rather than $d$, and the norm bound parameter is $2\Lambda$.
 6: **return** Private predictor $h_{w^{\mathsf{Priv}}} : \mathcal{X} \to \mathbb{R}$ defined as $\forall x \in \mathcal{X}, \ h_{w^{\mathsf{Priv}}}^{\widehat{\psi}}(x) \triangleq \big\langle w^{\mathsf{Priv}}, \widehat{\psi}(x) \big\rangle$.

---

*where, for any $h \in \mathcal{H}_\Lambda$, $\widehat{L}_S^\rho(h) \triangleq \frac{1}{m} \sum_{i=1}^m \ell^\rho \left( y_i \left\langle h, \psi(x_i) \right\rangle_{\mathbb{H}} \right)$, where $\psi$ is the feature map associated with the kernel $K$ and $\langle \cdot, \cdot \rangle_{\mathbb{H}}$ is the inner product associated with the RKHS $\mathbb{H}$.*

**Polynomial kernels:** Our results can be extended to polynomial kernels using a different approach to construct a finite-dimensional approximation of the kernel. A polynomial kernel of degree $p$, denoted as $\kappa_p$, can be expressed as $\kappa_p(x, x') = (\langle x, x' \rangle + c)^p$, $x, x' \in \mathcal{X}$ and $c > 0$ is some constant. Note that a feature map $\psi_p$ associated with such a kernel can be expressed as a vector in $\mathbb{R}^{\bar{d}}$, where $\bar{d} = O(d^p)$. In particular, $\psi_p(x)$ is the vector of all monomials of a $p$-th degree polynomial. Ignoring computational efficiency considerations (or when $p$ is a small constant), there is a simpler private construction than the one used for shift-invariant kernels. In that case, we can directly use the JL-transform to embed $\{\psi_p(x_1), \ldots, \psi_p(x_m)\}$ into a $k$-dimensional subspace exactly as in Section 3.2, which would result in a $k$-dimensional approximation of the kernel (by the properties of the JL-transform). Hence, we can directly use Algorithm 2 on the dataset $S_{\psi_p} \triangleq ((\psi_p(x_1), y_1), \ldots, (\psi_p(x_m), y_m))$. We therefore obtain the same bound on the expected error as above except that $r$ would then be $r^p$. That dependence on $r^p$ is inherent in this case even non-privately since $\kappa_p(x, x)$ can be as large as $r^p$. However, as discussed below, more efficient solutions can be designed for approximating these and many other kernels.

**Further extensions.** Our work can directly benefit from the method of [LSS13], which is computationally faster than that of [RR07], $O((m+d) \log d)$, instead of $O(md)$. Their technique also extends to any kernel that is a function of an inner product in the input space. We can further use, instead of the JL-transform, the *oblivious sketching* technique of [AKK+20] from numerical linear algebra, which builds on previous work by [PP13] and [ANW14], to design sketches for polynomial kernel with a target dimension that is only polynomially dependent on the degree of the kernel function, as well as a sketch for the Gaussian kernel on bounded datasets that does not suffer from an exponential dependence on the dimensionality of input data points. More recently, [SWYZ21] presented new oblivious sketches that further considerably improved upon the running-time complexity of these techniques. Their method also applies to other *slow-growing* kernels such as the neural tangent (NTK) and arc-cosine kernels.

# 5 Private Algorithms for Learning Neural Networks with Margin Guarantees

In this section, we describe a private learning algorithm that benefits from favorable margin guarantees when run with a family of neural networks with a very large input dimension.

We consider a family $\mathcal{H}_{\mathsf{NN}}$ of $L$-layer feed-forward neural networks defined over $\mathbb{B}^d(r)$, with $d$ potentially very large compared to the sample size $m$. A function $h$ in $\mathcal{H}_{\mathsf{NN}}$ can be viewed as a cascade of linear maps composed with a non-linear activation function (see Figure 2, left column). Here, $W_1, \ldots, W_L$ are the weight matrices defining the network and $\psi$ is a non-linear activation. For simplicity, the width (number of neurons) in each hidden layer, denoted by $N$, is assumed to be the same for all the layers. Also, we assume that the output of the network is a

real scalar and hence we have $W_L \in \mathbb{R}^N$. Furthermore, we assume no activation in the output layer. We also assume the same activation $\psi \colon \mathbb{R}^N \to \mathbb{R}^N$ for all layers and choose it to be a sigmoid function: for any $u = (u_1, \ldots, u_N) \in \mathbb{R}^N$, $\psi(u) = (\sigma_\eta(u_1), \ldots, \sigma_\eta(u_N))$, for some $\eta > 0$, where $\sigma_\eta(a) = \frac{1 - e^{-\frac{\eta a}{2}}}{1 + e^{-\frac{\eta a}{2}}}$, $a \in \mathbb{R}$. Note that $\sigma_\eta$ is $\eta$-Lipschitz and thus $\psi$ is $\eta$-Lipschitz with respect to $\|\cdot\|_2$: for any $u, v \in \mathbb{R}^N$, $\|\psi(u) - \psi(v)\|_2 \leq \eta \|u - v\|_2$. A typical choice for $\eta$ in practice is $\eta = 1$, but we will keep the dependence on $\eta$ for generality. We define $\mathcal{H}_{\mathsf{NN}^\Lambda}$ as the subset of $\mathcal{H}_{\mathsf{NN}}$ with weight matrices that are $\Lambda$-bounded in their Frobenius norm: for all $j \in [L]$, $\|W_j\|_F \leq \Lambda$ for some $\Lambda > 0$.

We design a pure DP algorithm for learning $L$-layer feed-forward networks in $\mathcal{H}_{\mathsf{NN}^\Lambda}$ that benefits from the following margin-based guarantee.

**Theorem 5.1.** *Let $\varepsilon > 0, \beta \in (0, 1)$, and $\rho > 0$. Then, there is an $\varepsilon$-DP algorithm which returns an $L$-layer network $h^{\mathsf{Priv}}$ with $N$ neurons per layer that with probability at least $1 - \beta$ over the draw of a sample $S \sim \mathcal{D}^m$ and the internal randomness of the algorithm admits the following guarantee:*

$$R_{\mathcal{D}}(h^{\mathsf{Priv}}) \leq \min_{h \in \mathcal{H}_{\mathsf{NN}^\Lambda}} \widehat{R}_S^\rho(h) + O\left( \frac{r(2\eta\Lambda)^L \sqrt{N\theta}}{\rho\sqrt{m}} + \frac{r^2(2\eta\Lambda)^{2L} N\theta}{\rho^2 \varepsilon m} \right),$$

*where $\theta = \log(Lm/\beta) \log(r(\eta\Lambda)^L/\rho)$.*

Note that this guarantee is independent of $d$ and, assuming $L$ is a constant, the bound scales roughly as $\sqrt{\frac{N}{\rho^2 m}} + \frac{N}{\rho^2 \varepsilon m}$, where $\rho$ is the confidence-margin parameter and $\varepsilon$ is the privacy parameter. Note that, for $\varepsilon \approx 1$, the bound scales with $\sqrt{\#\text{ neurons}}$, which is more favorable than standard bounds obtained via a uniform convergence argument, which depend on $d$, as well as the total number of edges $\Omega(N^2)$, in addition to a similar dependence on $\Lambda^L$.

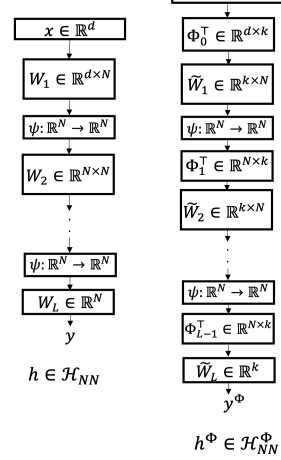

**Figure 2:** Illustration of the neural networks before and after JL-transforms.

**Our construction.** Our DP learner is based on using $L$ embeddings $\Phi_0 \in \mathbb{R}^{k \times d}, \ldots, \Phi_{L-1} \in \mathbb{R}^{k \times N}$ given by data-independent JL-transform matrices to reduce the dimension of the inputs in each layer, including the input layer, to $k = O\left( \frac{r^2(2\eta\Lambda)^{2L}}{\rho^2} \right)$. We randomly generate a set $\Phi = (\Phi_0, \ldots, \Phi_{L-1})$ of $L$ independent JL matrices whose dimensions are described as above. We let $\mathcal{H}_{\mathsf{NN}}^\Phi$ denote the family of $L$-layer neural networks, where each network $h^\Phi \in \mathcal{H}_{\mathsf{NN}}^\Phi$ is associated with weight matrices $\Phi_0^\top \widetilde{W}_1, \ldots, \Phi_{L-1}^\top \widetilde{W}_L$ for $\widetilde{W}_j \in \mathbb{R}^{k \times N}$, $j \in [L-1]$, and $\widetilde{W}_L \in \mathbb{R}^{k \times 1}$ (see Figure 2, right column). We define $\mathcal{H}_{\mathsf{NN}^{2\Lambda}}^\Phi \subset \mathcal{H}_{\mathsf{NN}}^\Phi$ where $\|\widetilde{W}_j\|_F \leq 2\Lambda$ for all $j \in [L]$. We start by creating a $\gamma$-cover $\mathcal{C}$ of the product space of the matrices $\widetilde{W}_1, \ldots, \widetilde{W}_L$ associated with $\mathcal{H}_{\mathsf{NN}^{2\Lambda}}^\Phi$, where $\gamma = O\left( \frac{\rho}{r(4\eta\Lambda)^L} \right)$. $\mathcal{C}$ is a $\gamma$-cover of $\mathbb{B}^{k \times N}(2\Lambda) \times \ldots \times \mathbb{B}^{k \times N}(2\Lambda) \times \mathbb{B}^k(2\Lambda)$ with respect to $\sqrt{\sum_{j=1}^L \|\cdot\|_F^2}$. We define $\widehat{\mathcal{H}}_{\mathsf{NN}^{2\Lambda}}^\Phi \subset \mathcal{H}_{\mathsf{NN}^{2\Lambda}}^\Phi$ to be the corresponding family of networks whose associated matrices are in $\mathcal{C}$. Given an input dataset $S = ((x_1, y_1), \ldots, (x_m, y_m)) \in \left( \mathbb{B}^d(r) \times \{\pm 1\} \right)^m$, we then run the exponential mechanism over $S$ with privacy parameter $\varepsilon$, the score function being the empirical zero-one loss $\widehat{R}_S(h) \colon h \in \widehat{\mathcal{H}}_{\mathsf{NN}^{2\Lambda}}^\Phi$, and the sensitivity $1/m$, to return a neural network $h^{\mathsf{Priv}} \in \widehat{\mathcal{H}}_{\mathsf{NN}^{2\Lambda}}^\Phi$.

## 6 Conclusion

We presented a series of new differentially private algorithms with dimension-independent margin guarantees, including algorithms for linear classification, kernel-based classification, or learning with a family of feed-forward neural networks, and label DP learning with general hypothesis sets. Our kernel-based algorithms can be extended to non-linear classification with many other kernels, including a variety of kernels that can be approximated using polynomial kernels, using techniques based on oblivious sketching. Our study of DP algorithms with margin guarantees for a family of neural networks can be viewed as an initiatory step that could serve as the basis for a more extensive analysis of DP algorithms for broader families of neural networks.

## Acknowledgments and Disclosure of Funding

This work was done while RB was a visiting scientist at Google, NY. RB's work at OSU on this research is supported by NSF Award AF-1908281 and Google Faculty Research Award.

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
