# Contents of Appendix

# A Useful Lemmas

We use empirical Bernstein bounds, properties of exponential mechanism and Johnson-Lindenstrauss lemmas which we state below.

**Lemma A.1** (Relative deviation bound ). *For any hypothesis set $\mathcal{H}$ of functions mapping from $\mathcal{X}$ to $R$, with probability at least $1 - \beta$, the following inequality holds for all $h \in \mathcal{H}$:*

$$R_{\mathcal{D}}(h) \leq \widehat{R}_S(h) + 2\sqrt{\widehat{R}_S(h)\frac{V(\mathcal{H})\log(2m) + \log\frac{4}{\beta}}{m}} + 4\frac{V(\mathcal{H})\log(2m) + \log\frac{4}{\beta}}{m},$$

*where $V(\mathcal{H})$ is the VC-dimension of class $\mathcal{H}$.*

The above lemma is obtained by combining [CGM19, Corollary 7] and VC-dimension bounds.

**Lemma A.2** (Relative deviation margin bound [CMTS21]). *Fix $\rho \geq 0$. Then, for any hypothesis set $\mathcal{H}$ of functions mapping from $\mathcal{X}$ to $\mathbb{R}$ with $d = \mathrm{fat}_{\frac{\rho}{16}}(\mathcal{H})$, with probability at least $1 - \beta$, the following holds for all $h \in \mathcal{H}$:*

$$R_{\mathcal{D}}(h) \leq \widehat{R}_S^\rho(h) + 2\sqrt{\widehat{R}_S^\rho(h)\frac{M}{m}} + \frac{M}{m},$$

*where $M = 1 + d\log_2(2c^2 m)\log_2\frac{2cem}{d} + \log\frac{1}{\beta}$ and $c = 17$.*

**Lemma A.3.** *Let $\beta, \gamma \in (0, 1)$. Let $T \subset \mathbb{R}^d$ be any set of $m$ vectors. There exists $k = O\left(\frac{\log\left(\frac{m}{\beta}\right)}{\gamma^2}\right)$ such that for any random $k \times d$ matrix $\Phi$ with entries drawn i.i.d. uniformly from $\{\pm\frac{1}{\sqrt{k}}\}$, the following inequalities hold simultaneously with probability at least $1 - \beta$ over the choice of $\Phi$:*

- *For any $u \in T$,*

$$\left(1 - \frac{\gamma}{3}\right)\|u\|_2^2 \leq \|\Phi u\|_2^2 \leq \left(1 + \frac{\gamma}{3}\right)\|u\|_2^2.$$

- *For any $u, v \in T$,*

$$|\langle \Phi u, \Phi v\rangle - \langle u, v\rangle| \leq \frac{\gamma}{3}\|u\|_2\|v\|_2.$$

*Proof.* The first property is simply the Johnson-Lindenstrauss (JL) property and follows from the standard JL lemma (see, e.g., [JL84, LN17]). Below we show both first and second property holds simultaneously. Define

$$\widetilde{T} \triangleq \left\{z \in \mathbb{R}^d : z = \frac{u}{\|u\|_2} \pm \frac{v}{\|v\|_2}, \ u, v \in T\right\}.$$

Note that the number of non-zero vectors in $\widetilde{T}$ is at most $m^2$. By the JL lemma [JL84, LN17] over the set $T \cup \widetilde{T}$, there exists $k = O\left(\frac{\log(m^2/\beta)}{\gamma^2}\right) = O\left(\frac{\log\left(\frac{m}{\beta}\right)}{\gamma^2}\right)$ and $\Phi'$ such that with probability $\geq 1 - \beta$, for all $u, v \in T$ we have

$$\left(1 - \frac{\gamma}{3}\right)\|\bar{u} + \bar{v}\|_2^2 \leq \|\Phi'(\bar{u} + \bar{v})\|_2^2 \leq \left(1 + \frac{\gamma}{3}\right)\|\bar{u} + \bar{v}\|_2^2, \tag{3}$$

$$\left(1 - \frac{\gamma}{3}\right)\|\bar{u} - \bar{v}\|_2^2 \leq \|\Phi'(\bar{u} - \bar{v})\|_2^2 \leq \left(1 + \frac{\gamma}{3}\right)\|\bar{u} - \bar{v}\|_2^2, \tag{4}$$

$$\left(1 - \frac{\gamma}{3}\right)\|u\|_2^2 \leq \|\Phi'u\|_2^2 \leq \left(1 + \frac{\gamma}{3}\right)\|u\|_2^2. \tag{5}$$

where $\bar{u} = \frac{u}{\|u\|_2}$ and $\bar{v} = \frac{v}{\|v\|_2}$. (5) implies the first result in the lemma. Now, fix any $u, v \in T$. Let $\bar{u} = \frac{u}{\|u\|_2}$ and $\bar{v} = \frac{v}{\|v\|_2}$. Observe that for any $\mathbf{a}, \mathbf{b} \in \mathbb{R}^d$, we have $\langle \mathbf{a}, \mathbf{b}\rangle = \frac{1}{4}\left(\|\mathbf{a} + \mathbf{b}\|_2^2 - \|\mathbf{a} - \mathbf{b}\|_2^2\right)$.

Hence, we have

$$\left|\langle \bar{u}, \bar{v}\rangle - \langle \Phi'\bar{u}, \Phi'\bar{v}\rangle\right| \le \frac{1}{4}\left|\|\bar{u}+\bar{v}\|_2^2 - \|\Phi'(\bar{u}+\bar{v})\|_2^2\right| + \frac{1}{4}\left|\|\Phi'(\bar{u}-\bar{v})\|_2^2 - \|\bar{u}-\bar{v}\|_2^2\right| \qquad (\ )$$

$$\le \frac{\gamma}{12}\left(\|\bar{u}+\bar{v}\|_2^2 + \|\bar{u}-\bar{v}\|_2^2\right)$$

$$\le \frac{\gamma}{3},$$

where the second inequality follows from (3) and (4) 'and the third inequality follows from the triangle inequality and the fact that $\|\bar{u}\|_2 = \|\bar{v}\|_2 = 1$. Hence, we finally have

$$|\langle u, v\rangle - \langle \Phi'u, \Phi'v\rangle| \le \|u\|_2\|v\|_2 \left|\langle \bar{u}, \bar{v}\rangle - \langle \Phi'\bar{u}, \Phi'\bar{v}\rangle\right| \le \frac{\gamma}{3}\|u\|_2\|v\|_2.$$

$\square$

The time complexity to apply the random matrix $\Phi$ in Lemma A.3 to a vector $v$ is $k \cdot d$, which can be prohibitive in many cases. There are several works which provide $\Phi$ that support fast matrix vector products. [AC06] provides a $\Phi$ which can be applied in time $c'd\log d + \frac{c'\log(d/\beta)\log^2(1/\beta)}{\gamma^2}$, however the results are stated with constant probability. [Nel10] gave a slightly different construction which can be applied in time $c'd\log d + \frac{c'\log(d/\beta)\log^2(1/\beta)}{\gamma^4}$ and the results hold with high probability. [AL09] provided a construction which can be applied in time $c'd\log k$ for $k = O(d^{0.499})$. [KW11] showed that any RIP matrix can be used for JL-transform and provided JL-transform results for several fast random projections. Since we need high probability bounds without any restrictions, we use the following result, which is computationally efficient, but is suboptimal in the projection dimension up to logarithmic factors.

**Lemma A.4.** *Let $\beta, \gamma \in (0, 1)$ and $c$ and $c'$ be sufficiently large constants. Let $T \subset \mathbb{R}^d$ be any set of $m$ vectors. Let $k = \frac{c\log\left(\frac{m}{\beta}\right)\log\left(\frac{m}{\gamma\beta}\right)}{\gamma^2}$. There exists a matrix $\Phi$ which can be applied to any vector $v$ in time $c'd\log d + c'k$, such that the following inequalities hold simultaneously with probability at least $1 - \beta$ over the choice of $\Phi$:*

- *For any $u \in T$,*

$$\left(1 - \frac{\gamma}{3}\right)\|u\|_2^2 \le \|\Phi u\|_2^2 \le \left(1 + \frac{\gamma}{3}\right)\|u\|_2^2. \qquad (6)$$

- *For any $u, v \in T$,*

$$|\langle \Phi u, \Phi v\rangle - \langle u, v\rangle| \le \frac{\gamma}{3}\|u\|_2\|v\|_2. \qquad (7)$$

Property (6) in the above result follows directly from [Nel15] and the proof for property (7) is similar to that of Lemma A.3 and is omitted.

# B  DP Algorithms for Linear Classification with Margin Guarantees

## B.1  Proof of Theorem 3.1

**Theorem 3.1.** *Algorithm 1 is $\varepsilon$-differentially private. For any $\beta \in (0, 1)$, with probability at least $1 - \beta$ over the draw of a sample $S$ of size $m$ from $\mathcal{D}$, the solution $w^{\mathsf{Priv}}$ it returns satisfies:*

$$R_{\mathcal{D}}(w^{\mathsf{Priv}}) \le \min_{w \in \mathbb{B}^d(\Lambda)}\left\{\widehat{R}_S^\rho(w) + \widetilde{O}\left(\sqrt{\widehat{R}_S^\rho(w)\frac{\Lambda^2 r^2}{m\rho^2}} + \frac{\Lambda^2 r^2}{\rho^2\min(1,\varepsilon)m}\right)\right\}.$$

A more precise version of the above bound is given as follows:

$$R_{\mathcal{D}}(w^{\mathsf{Priv}}) \le \min_{w \in \mathbb{B}^d(\Lambda)}\left\{\widehat{R}_S^\rho(w) + O\left(\sqrt{\widehat{R}_S^\rho(w)\left(\frac{\Lambda^2 r^2\log^2\left(\frac{m}{\beta}\right)}{m\rho^2} + \frac{\log\left(\frac{1}{\beta}\right)}{m}\right)} + \Gamma\right)\right\},$$

$$\text{where} \quad \Gamma = \frac{\Lambda^2 r^2\log\left(\frac{m}{\beta}\right)\log\left(\frac{\Lambda r}{\beta\rho}\right)}{\rho^2\varepsilon m} + \frac{\Lambda^2 r^2\log^2\left(\frac{m}{\beta}\right)}{m\rho^2} + \frac{\log\left(\frac{1}{\beta}\right)}{m}.$$

*Proof.* The proof of privacy follows from combining the following two properties: $\tilde{w}$ is generated via the exponential mechanism, which an $\varepsilon$-differentially private mechanism, and $\Phi$ is generated independently of $S$.

We now prove the accuracy guarantee of Algorithm 1. If $m < \frac{c\Lambda^2 r^2 \log\left(\frac{m}{\beta}\right) \log\left(\frac{20\Lambda r}{\beta\rho}\right)}{\rho^2 \varepsilon}$, for some constant $c$, the bound follow trivially. Hence in the rest of the proof we assume that $m$ is at least $\frac{c\Lambda^2 r^2 \log\left(\frac{m}{\beta}\right) \log\left(\frac{20\Lambda r}{\beta\rho}\right)}{\rho^2 \varepsilon}$ for some large constant $c$. Let

$$\alpha = \frac{c\Lambda^2 r^2 \log\left(\frac{m}{\beta}\right) \log\left(\frac{20\Lambda r}{\beta\rho}\right)}{\rho^2 \varepsilon m}.$$

First, observe that

$$\begin{aligned}
\widehat{R}_S(w^{\mathsf{Priv}}) &= \frac{1}{m} \sum_{(x,y)\in S} \mathbf{1}\left(y\langle w^{\mathsf{Priv}}, x\rangle\right) \\
&= \frac{1}{m} \sum_{(x,y)\in S} \mathbf{1}\left(y\langle \Phi^{\top}\tilde{w}, x\rangle\right) \\
&= \frac{1}{m} \sum_{(x,y)\in S} \mathbf{1}\left(y\langle \tilde{w}, \Phi x\rangle\right) \\
&= \frac{1}{m} \sum_{(x_\Phi,y)\in S_\Phi} \mathbf{1}\left(y\langle \tilde{w}, x_\Phi\rangle\right) \\
&= \widehat{R}_{S_\Phi}^{(k)}(\tilde{w}).
\end{aligned}$$

Let $\widehat{w} \in \underset{w\in\mathcal{C}}{\operatorname{argmin}}\, \widehat{R}_{S_\Phi}^{(k)}(w)$. Note that $|\mathcal{C}| = \left(\frac{20\Lambda r}{\rho}\right)^k$, where $k = \frac{c'\Lambda^2 r^2 \log\left(\frac{m}{\beta}\right)}{\rho^2}$. Hence, by the accuracy properties of the exponential mechanism and the fact that $m \geq \frac{4\log\left(\frac{4|\mathcal{C}|}{\beta}\right)}{\varepsilon\alpha}$, we have that with probability at least $1 - \beta/4$,

$$\widehat{R}_{S_\Phi}^{(k)}(\tilde{w}) \leq \widehat{R}_{S_\Phi}^{(k)}(\widehat{w}) + \alpha.$$

Combining the above facts, we get that with probability at least $1 - \beta/4$,

$$\widehat{R}_S(w^{\mathsf{Priv}}) \leq \widehat{R}_{S_\Phi}^{(k)}(\widehat{w}) + \alpha. \tag{8}$$

Let $w^* \in \underset{w\in\mathbb{B}^d(\Lambda)}{\operatorname{argmin}}\, \widehat{R}_S^{\rho}(w)$ and let $w_\Phi^* = \Phi w^*$. Note that

$$\|w_\Phi^*\|_2 \leq \sqrt{1 + \frac{\rho}{3\Lambda r}} \|w^*\|_2 \leq 2\|w^*\|_2 \leq 2\Lambda,$$

and hence $w_\Phi^* \in \mathbb{B}^k(2\Lambda)$. Since $\mathcal{C}$ is $\frac{\rho}{10 r}$-cover of $\mathbb{B}^k(2\Lambda)$, then there must be $w_c \in \mathcal{C}$ such that $\|w_c - w_\Phi^*\|_2 \leq \frac{\rho}{10 r}$. Hence, observe that for any $(x_\Phi, y) \in S_\Phi$,

$$\begin{aligned}
y\langle w_c, x_\Phi\rangle &= y\langle w_\Phi^*, x_\Phi\rangle + y\langle w_c - w_\Phi^*, x_\Phi\rangle \\
&\geq y\langle w_\Phi^*, x_\Phi\rangle - \|w_c - w_\Phi^*\|_2 \|x_\Phi\|_2 \\
&\geq y\langle w_\Phi^*, x_\Phi\rangle - \frac{\rho}{10 r}\|x_\Phi\|_2.
\end{aligned}$$

Now, by Lemma A.3, with probability at least $1 - \beta/4$, for all $x_\Phi$ s.t. $(x_\Phi, y) \in S_\Phi$ we have $\|x_\Phi\|_2 \leq \sqrt{1 + \frac{\rho}{3\Lambda r}}\|x\|_2 \leq \sqrt{1 + \frac{\rho}{3\Lambda r}}r$. Hence, we get that with probability at least $1 - \beta/4$ for all $(x_\Phi, y) \in S_\Phi$

$$y\langle w_c, x_\Phi\rangle \geq y\langle w_\Phi^*, x_\Phi\rangle - 0.15\rho.$$

The last inequality implies that for any $\rho' > 0$, with probability at least $1 - \beta/4$ (over the choice of $\Phi$), we must have $\widehat{R}_{S_\Phi}^{\rho',(k)}(w_c) \leq \widehat{R}_{S_\Phi}^{\rho'+0.15\rho,(k)}(w_\Phi^*)$. In particular, with probability at least $1 - \beta/4$ we have

$$\widehat{R}_{S_\Phi}^{0.5\rho,(k)}(w_c) \leq \widehat{R}_{S_\Phi}^{0.65\rho,(k)}(w_\Phi^*). \tag{9}$$

Moreover, by the definition of $\widehat{w}$, we have $\widehat{R}_{S_\Phi}^{(k)}(\widehat{w}) \le \widehat{R}_{S_\Phi}^{(k)}(w_c) \le \widehat{R}_{S_\Phi}^{0.5\rho,(k)}(w_c)$. Combining this fact with (8) and (9), we get that with probability at least $1 - \beta/2$

$$\widehat{R}_S(w^{\mathsf{Priv}}) \le \widehat{R}_{S_\Phi}^{0.65\rho,(k)}(w_\Phi^*) + \alpha. \tag{10}$$

Now, by Lemma A.3 and the fact that $\|w^*\|_2 \le \Lambda$ and $\|x\|_2 \le r$, it follows that with probability at least $1 - \beta/4$ for all $(x_\Phi, y) \in S_\Phi$, we have

$$y\langle w^*, x\rangle \ge \rho \implies y\langle w_\Phi^*, x_\Phi\rangle \ge \rho/3.$$

This directly implies that with probability at least $1 - \beta/4$,

$$\widehat{R}_{S_\Phi}^{0.65\rho,(k)}(w_\Phi^*) \le \widehat{R}_S^\rho(w^*).$$

Combining this with (10), we can assert that with probability at least $1 - \frac{3}{4}\beta$, we have

$$\widehat{R}_S(w^{\mathsf{Priv}}) \le \widehat{R}_S^\rho(w^*) + \alpha. \tag{11}$$

In the final step of the proof, we rely on a standard uniform convergence argument to bound $R_{\mathcal{D}}(w^{\mathsf{Priv}})$ in terms of $\widehat{R}_S(w^{\mathsf{Priv}})$. Note that the VC-dimension of $\{\mathrm{sgn} \circ h_w : w \in \mathcal{C}\}$ is $k$. By Lemma A.1, with probability at least $1 - \beta/4$

$$R_{\mathcal{D}}(w^{\mathsf{Priv}}) - \widehat{R}_S(w^{\mathsf{Priv}}) \le 2\sqrt{\widehat{R}_S(w^{\mathsf{Priv}})\frac{k\log(2m) + \log(16/\beta)}{m}} + 4\frac{k\log(2m) + \log(16/\beta)}{m}.$$

Combining the above two equations, we get with probability at least $1 - \beta$,

$$R_{\mathcal{D}}(w^{\mathsf{Priv}}) \le \widehat{R}_S^\rho(w^*) + 2\sqrt{\widehat{R}_S^\rho(w^*)\frac{k\log(2m) + \log(16/\beta)}{m}} + 2\alpha + 8\frac{k\log(2m) + \log(16/\beta)}{m}.$$

The lemma follows from observing that $w^* \in \underset{w\in\mathbb{B}^d(\Lambda)}{\mathrm{argmin}}\, \widehat{R}_S^\rho(w)$ if and only if $w^* \in \underset{w\in\mathbb{B}^d(\Lambda)}{\mathrm{argmin}}\, \widehat{R}_S^\rho(w) +$

$2\sqrt{\widehat{R}_S^\rho(w)\frac{k\log(2m)+\log(16/\beta)}{m}}$. $\hfill\square$

## B.2   Algorithm 4 of Section 3.2 and Proof of Lemma 3.1

Here, we give the details of Algorithm 4 invoked in step 4 of Algorithm 2. We describe here a more general setup where the loss function is any (possibly non-smooth) convex generalized linear loss (GLL). Given a parameter space $\mathcal{W}$, feature space $\mathcal{X}$, and label/target set $\mathcal{Y}$, a GLL is a loss function defined over $\mathcal{W} \times (\mathcal{X} \times \mathcal{Y})$ that can be written as $\ell(\langle w, x\rangle, y)$, $w \in \mathcal{W}, x \in \mathcal{X}, y \in \mathcal{Y}$ for some function $\ell : \mathbb{R} \times \mathcal{Y} \to \mathbb{R}$. Here, we assume that for any $y \in \mathcal{Y}$, $\ell(\cdot, y)$ is convex and $\frac{1}{\rho}$-Lipschitz. We also assume that $\mathcal{W} \subseteq \mathbb{B}^k(\Lambda)$ for some $\Lambda > 0$, $\mathcal{X} \subseteq \mathbb{B}^k(\tilde{r})$ for some $\tilde{r} > 0$, and $\mathcal{Y} \subseteq [-1, 1]$. Given a dataset $\widetilde{S} = ((x_1, y_1), \ldots, (x_m, y_m)) \in (\mathcal{X} \times \mathcal{Y})^m$, we define the empirical risk of $w \in \mathcal{W}$ with respect to $\widetilde{S}$ as $\widehat{L}_{\widetilde{S}}(w) \triangleq \frac{1}{m}\sum_{i=1}^m \ell(\langle w, x_i\rangle, y)$. Note that the setup in Algorithm 2 is a special case of the above.

Given an input dataset $\widetilde{S} \in (\mathcal{X} \times \mathcal{Y})^m$, Algorithm 4 below invokes the "Phased SGD algorithm for GLL" [BGM21, Algorithm 2] on a set $\widehat{S}$ of $m$ samples drawn uniformly with replacement from $\widetilde{S}$, and hence obtain an output $\tilde{w} \in \mathcal{W}$. In the sequel, we will refer to the algorithm in [BGM21] as $\mathcal{A}_{\mathsf{GLL}}$. Note that the expected loss with respect to the choice of $\widehat{S} \leftarrow \widetilde{S}$ is the empirical risk with respect to $\widetilde{S}$. Hence, one can derive a bound on the expected excess empirical risk that is roughly the same as the bound in [BGM21, Theorem 6] on the expected excess population risk. However, we note that ensuring that Algorithm 4 is $(\varepsilon, \delta)$-DP does not follow directly from the privacy guarantee of the $\mathcal{A}_{\mathsf{GLL}}$ since the sample $\widehat{S}$ may contain duplicate entries from $\widetilde{S}$. Nonetheless, we show that the privacy guarantee can be attained by appropriately setting the input privacy parameters to $\mathcal{A}_{\mathsf{GLL}}$ together with a careful privacy analysis. To transform the in-expectation bound into a high-probability bound, we perform a standard confidence-boosting procedure [BST14, Appendix D], where the procedure described above is repeated independently $M = O(\log(1/\beta))$ times to generate $\tilde{w}^1, \ldots, \tilde{w}^M$, and finally, the exponential mechanism (with a score function $-\widehat{L}_{\widetilde{S}}$) is used to privately select a final output $\tilde{w} \in \{\tilde{w}^1, \ldots, \tilde{w}^M\}$.

**Algorithm 4** DP-ERM algorithm for GLLs

---

**Require:** Private dataset $\widetilde{S} = \big((x_1, y_1), \ldots, (x_m, y_m)\big) \in (\mathcal{X} \times \mathcal{Y})^m$, where $\mathcal{X} \subseteq \mathbb{B}^k(\tilde{r})$ and $\mathcal{Y} \subseteq [-1, 1]$; parameter space $\mathcal{W} \subseteq \mathbb{B}^k(\Lambda)$; privacy parameters $(\varepsilon, \delta)$; confidence parameter $\beta \in (0, 1)$; convex, $\frac{1}{\rho}$-Lipschitz loss function $\ell$ for some $\rho > 0$; Oracle access to algorithm $\mathcal{A}_{\mathsf{GLL}}$ [BGM21, Algorithm 2].
1: Let $M := \log(2/\beta)$.
2: Let $\varepsilon' := \frac{\varepsilon}{4M \log(2M/\delta)}$.
3: Let $\delta' := \frac{\delta^2}{4M \log(2M/\delta)}$.
4: **for** $t = 1$ to $M$ **do**
5:  Sample $\widehat{S}^t = \big((\widehat{x}_1^t, \widehat{y}_1^t), \ldots (\widehat{x}_m^t, \widehat{y}_m^t)\big) \leftarrow \widetilde{S}$ uniformly with replacement.
6:  $\tilde{w}^t = \mathcal{A}_{\mathsf{GLL}}(\widehat{S}^t; \varepsilon', \delta')$, where $\mathcal{A}_{\mathsf{GLL}}$ is [BGM21, Algorithm 2] (the other obvious inputs to $\mathcal{A}_{\mathsf{GLL}}$ are omitted; the smoothing parameter and the oracle accuracy parameter of $\mathcal{A}_{\mathsf{GLL}}$ are set as in [BGM21, Theorem 6]).
7: Run the exponential mechanism with privacy parameter $\varepsilon/2$ to select $\tilde{w}$ from the set $\big(\tilde{w}^1, \ldots, \tilde{w}^M\big)$ associated with scores $\big(-\widehat{L}_{\widetilde{S}}(\tilde{w}^t) : t \in [M]\big)$.
8: **return** $\tilde{w}$

---

**Lemma 3.1.** *Let $m \in \mathbb{N}$, $0 < \delta < \frac{1}{m}$, and $0 < \varepsilon \leq \log(1/\delta)$. Algorithm 4 (Appendix B.2) is $(\varepsilon, \delta)$-DP. Let $\beta \in (0, 1)$. Let $k \in \mathbb{N}$, and $\tilde{r}, \Lambda > 0$. Let $\widetilde{S} \in \big(\mathbb{B}^k(\tilde{r}) \times \{\pm 1\}\big)^m$ be the input dataset and $\mathbb{B}^k(\Lambda)$ be the parameter space. With probability $1 - \beta$ over the randomness in Algorithm 4, the output $\tilde{w}$ satisfies*

$$\widehat{L}_{\widetilde{S}}^\rho(\tilde{w}) \leq \min_{w \in \mathbb{B}^k(\Lambda)} \widehat{L}_{\widetilde{S}}^\rho(w) + \frac{\Lambda \tilde{r}}{\rho} \cdot O\left(\frac{1}{\sqrt{m}} + \frac{\sqrt{k} \log^{\frac{3}{2}}(\frac{1}{\delta}) \log(\frac{1}{\beta})}{\varepsilon m}\right).$$

*Moreover, Algorithm 2 requires $O(m \log(m) \log(1/\beta))$ gradient computations.*

*Proof.* First, we show the privacy guarantee. Fix a round $t \in [M]$ of Algorithm 4. We will show that the $t$-th round is $(\frac{\varepsilon}{2M}, \frac{\delta}{M})$-DP. Suppose we can do that. Then, by the basic composition property of DP, the entire $M$ rounds of the algorithm is $(\frac{\varepsilon}{2}, \delta)$-DP. Next, we note that step 7 is $(\frac{\varepsilon}{2}, 0)$-DP by the privacy guarantee of the exponential mechanism. Hence, again by basic composition of DP, we conclude that Algorithm 4 is $(\varepsilon, \delta)$-DP. Thus, it remains to show that for any fixed $t \in [M]$, the $t$-th round is $(\hat{\varepsilon}, \hat{\delta})$-DP, where $\hat{\varepsilon} = \frac{\varepsilon}{2M}$ and $\hat{\delta} = \frac{\delta}{M}$. Fix any data point $(x_i, y_i) \in \widetilde{S}$. Let $J$ denote the number of appearances of $(x_i, y_i)$ in $\widehat{S}^t$. Note that $J \sim \mathsf{Bin}(m, 1/m)$. Hence, using the multiplicative Chernoff's bound, $J \leq 2 \log(2/\hat{\delta})$ with probability at least $1 - \hat{\delta}/2$. We will show that conditioned on this event, the $t$-th round is $(\hat{\varepsilon}, \hat{\delta}/2)$-DP, which suffices to prove our privacy claim for round $t$. Given that $J \leq 2 \log(2/\hat{\delta})$ and since $\mathcal{A}_{\mathsf{GLL}}$ is $\big(\frac{\hat{\varepsilon}}{2 \log(2/\hat{\delta})}, \frac{\delta \hat{\delta}}{4 \log(2/\hat{\delta})}\big)$-DP with respect to to its input dataset $\widehat{S}^t$, then by the group-privacy property of DP [DR14], round $t$ is $(\hat{\varepsilon}, \delta'')$-DP with respect to the dataset $\widetilde{S}$, where

$$\begin{aligned}
\delta'' &= \frac{\delta \hat{\delta}}{4 \log(2/\hat{\delta})} \cdot \sum_{j=0}^{2 \log(2/\hat{\delta})} e^{\frac{\hat{\varepsilon}}{2 \log(2/\hat{\delta})} j} \\
&= \frac{\delta \hat{\delta}}{4 \log(2/\hat{\delta})} \cdot \frac{e^{\hat{\varepsilon}} - 1}{e^{\frac{\hat{\varepsilon}}{2 \log(2/\hat{\delta})}} - 1} \\
&\leq \frac{\delta \hat{\delta}(e^{\hat{\varepsilon}} - 1)}{2\hat{\varepsilon}} \\
&\leq \frac{\hat{\delta}}{2},
\end{aligned}$$

where the third inequality follows from the fact that $e^{\frac{\hat{\varepsilon}}{2 \log(2/\hat{\delta})}} - 1 \geq \frac{\hat{\varepsilon}}{2 \log(2/\hat{\delta})}$, and the last step follows from the fact that $\frac{e^a - 1}{a}$ is increasing in $a > 0$ and the assumption that $\varepsilon \leq \log(1/\delta)$ (and hence

$\hat{\varepsilon} < \varepsilon \le \log(1/\delta))$. Hence, we have shown that any given round of the algorithm is $(\varepsilon, \hat{\delta})$-DP. This concludes the proof of the privacy guarantee.

We now prove the bound on the excess empirical risk. Fix any round $t$. Let $\widehat{\mathcal{D}}_{\widetilde{S}}$ denote the empirical distribution of $\widetilde{S}$. Note that $\widehat{S}^t \sim \widehat{\mathcal{D}}_{\widetilde{S}}^m$, i.e., $\widehat{S}^t$ is comprised of $m$ independent samples from $\widehat{\mathcal{D}}_{\widetilde{S}}$. Hence, $\mathbb{E}_{(x,y)\sim\widehat{\mathcal{D}}_{\widetilde{S}}} [\ell(\langle w, x \rangle, y)] = \widehat{L}_{\widetilde{S}}(w)$. Thus, by the excess risk guarantee of $\mathcal{A}_{\mathsf{GLL}}$ [BGM21, Theorem 6], we have

$$\mathbb{E}\left[\widehat{L}_{\widetilde{S}}^\rho(\tilde{w})\right] - \min_{w\in\mathbb{B}^k(\Lambda)} \widehat{L}_{\widetilde{S}}^\rho(w) = \frac{\Lambda\tilde{r}}{\rho} \cdot O\left(\frac{1}{\sqrt{m}} + \frac{\sqrt{k\log(\frac{1}{\delta'})}}{\varepsilon' m}\right)$$

$$= \frac{\Lambda\tilde{r}}{\rho} \cdot O\left(\frac{1}{\sqrt{m}} + \frac{\sqrt{k}\log^{\frac{3}{2}}(\frac{1}{\delta})\log(\frac{1}{\beta})}{\varepsilon m}\right),$$

where the expectation is with respect to the sampling step (step 5 of Algorithm 4) and the randomness in $\mathcal{A}_{\mathsf{GLL}}$. Note the last step follows from the setting of $\varepsilon'$ and $\delta'$ in Algorithm 4 and the fact that $\log(\log(1/\beta)/\delta) = O(\log(1/\delta))$, which follows from the assumption $\delta < 1/m$ (in the statement of the lemma) and $\log(1/\beta) < m$ (since the bound would be trivial otherwise). Given this expectation guarantee on the output of each round, the final selection step (step 7) returns a parameter $\tilde{w}$ that satisfies the bound above with probability at least $1 - \beta$. This can be shown by following the same argument in [BST14, Appendix D] while noting that the sensitivity of the score function $-\widehat{L}_{\widetilde{S}}$ is bounded by $\frac{\Lambda\tilde{r}}{m}$.

Finally, the running time of $\mathcal{A}_{\mathsf{GLL}}$, measured in terms of gradient computations, is $O(m\log(m))$ [BGM21, Theorem 6]. Hence, the gradient complexity of Algorithm 4 is bounded by $O(m\log(m)\log(1/\beta))$. $\qquad\square$

## B.3 Proof of Theorem 3.2

**Theorem 3.2.** *Let $0 < \delta < \frac{1}{m}$ and $0 < \varepsilon \le \log(1/\delta)$. Algorithm 2 is $(\varepsilon, \delta)$-DP. Let $\beta \in (0,1)$. Let $S \sim \mathcal{D}^m$ for a distribution $\mathcal{D}$ over $\mathbb{B}^d(r) \times \{\pm 1\}$. Algorithm 2 outputs $w^{\mathsf{Priv}} \in \mathbb{R}^d$ such that with probability at least $1 - \beta$, we have*

$$R_{\mathcal{D}}(w^{\mathsf{Priv}}) \le \min_{w\in\mathbb{B}^d(\Lambda)} \widehat{L}_S^\rho(w) + \widetilde{O}\left(\frac{\Lambda r}{\rho\sqrt{\min(1,\varepsilon)\,m}}\right).$$

*Moreover, Algorithm 2 runs in time $O\left(md\log(\max(d,m)) + \varepsilon m^2\log(m)/\log^{\frac{3}{2}}(1/\delta)\right)$.*

A more precise version of the above bound is given as follows:

$$R_{\mathcal{D}}(w^{\mathsf{Priv}}) \le \min_{w\in\mathbb{B}^d(\Lambda)} \widehat{L}_S^\rho(w) + O\left(\sqrt{\frac{\log(1/\beta)}{m}} + \frac{\Lambda r}{\rho}\left(\frac{1}{\sqrt{m}} + \frac{\sqrt{\log(\frac{m}{\beta})\log(\frac{1}{\beta})}\log^{\frac{3}{4}}(\frac{1}{\delta})}{\sqrt{\varepsilon m}}\right)\right).$$

*Proof.* The proof of privacy follows directly from the $(\varepsilon, \delta)$-DP guarantee of Algorithm 4 (step 4) and the fact that DP is closed under post-processing.

Next, we will prove the claimed margin bound. For simplicity and without loss of generality, we will set $\Lambda = 1$. For the general setting of $\Lambda$, the claimed bound follows by rescaling the parameter vectors in the proof.

Recall that $x_\Phi \triangleq \Phi x$. Let $w^* \in \operatorname*{argmin}_{w\in\mathbb{B}^d} \widehat{L}_S^\rho(w)$. Define $w_\Phi^* \triangleq \Phi w^*$. By Lemma A.4, there is

$\gamma = O\left(\sqrt{\frac{\log(\frac{m}{\beta})}{k}}\right) = O\left(\sqrt{\frac{\log(1/\beta)}{\varepsilon m}}\log^{\frac{3}{4}}(1/\delta)\right)$ such that with probability at least $1 - \beta/3$ over the

randomness of $\Phi$, for every feature vector $x$ in the training set $S$, we have

$$\|x_\Phi\|_2^2 \le \left(1 + \frac{\gamma}{3}\right)\|x\|_2^2 \tag{12}$$

$$1 - \frac{y\langle w_\Phi^*, x_\Phi\rangle}{\rho} \le 1 - \frac{y\langle w^*, x\rangle}{\rho} + \frac{r\gamma}{\rho} \tag{13}$$

We condition on this event for the remainder of the proof.

Note that (12) implies that

$$S_\Phi = \{(x_\Phi, y) : (x, y) \in S\};$$

that is, for all feature vectors $x$ in the dataset $S$, $x_\Phi \in \mathbb{B}^k(2r)$ (i.e., $\Pi_{\mathbb{B}^k(2r)}(x_\Phi) = x_\Phi$).

Let $\mathcal{D}_\Phi$ denote the distribution of the pair $(x_\Phi, y)$, where $(x, y) \sim \mathcal{D}$. Via a standard margin bound [MRT18, Theorems 5.8 & 5.10], with probability at least $1 - \beta/3$ over the choice of the training set $S$, we have

$$\forall w \in \mathbb{B}^k \quad R_{\mathcal{D}_\Phi}(w) \le \widehat{L}_{S_\Phi}^\rho(w) + \frac{4r}{\rho\sqrt{m}} + 2\sqrt{\frac{\log(6/\beta)}{m}}$$

It follows that with probability at least $1 - \beta/3$, we have

$$R_{\mathcal{D}_\Phi}(\tilde{w}) \le \widehat{L}_{S_\Phi}^\rho(\tilde{w}) + \frac{4r}{\rho\sqrt{m}} + 2\sqrt{\frac{\log(6/\beta)}{m}},$$

where $\tilde{w}$ is the output of step 4 of Algorithm 2. Moreover, note that

$$R_{\mathcal{D}}(w^{\mathsf{Priv}}) = \mathbb{P}_{(x,y)\sim\mathcal{D}}\left[y\langle w^{\mathsf{Priv}}, x\rangle \le 0\right]$$
$$= \mathbb{P}_{(x_\Phi,y)\sim\mathcal{D}_\Phi}\left[y\langle \tilde{w}, x_\Phi\rangle \le 0\right]$$
$$= R_{\mathcal{D}_\Phi}(\tilde{w})$$

Thus, we get that with probability at least $1 - \beta/3$,

$$R_{\mathcal{D}}(w^{\mathsf{Priv}}) \le \widehat{L}_{S_\Phi}^\rho(\tilde{w}) + \frac{4r}{\rho\sqrt{m}} + 2\sqrt{\frac{\log(6/\beta)}{m}}. \tag{14}$$

By Lemma 3.1, with probability at least $1 - \beta/3$ over the randomness of Algorithm 4 (step 4 of Algorithm 2), we have that

$$\widehat{L}_{S_\Phi}^\rho(\tilde{w}) \le \widehat{L}_{S_\Phi}^\rho(\widehat{w}) + \frac{\Lambda r}{\rho}\left(\frac{1}{\sqrt{m}} + \frac{\sqrt{\log(\frac{m}{\beta})\log(\frac{1}{\beta})}\log^{\frac{3}{4}}(\frac{1}{\delta})}{\sqrt{\varepsilon m}}\right), \tag{15}$$

where $\widehat{w} \in \underset{w\in\mathbb{B}^k}{\operatorname{argmin}} \widehat{L}_{S_\Phi}^\rho(w)$. Moreover, (13) implies

$$\widehat{L}_{S_\Phi}^\rho(w_\Phi^*) \le \widehat{L}_S^\rho(w^*) + \frac{r}{\rho}\cdot O\left(\sqrt{\frac{\log(1/\beta)}{\varepsilon m}}\log^{\frac{3}{4}}(1/\delta)\right)$$

Note that by definition of $\widehat{w}$, we have $\widehat{L}^\rho(\widehat{w}; S) \le \widehat{L}_{S_\Phi}^\rho(w_\Phi^*)$. Hence, we have

$$\widehat{L}_{S_\Phi}^\rho(\widehat{w}) \le \widehat{L}_S^\rho(w^*) + \frac{r}{\rho}O\left(\sqrt{\frac{\log(1/\beta)}{\varepsilon m}}\log^{\frac{3}{4}}(1/\delta)\right) \tag{16}$$

Now, by combining (14), (15), and (16), we reach the desired bound.

Finally, concerning the running time, observe that the Fast JL-transform (steps 2 and 3) takes $O(md\log(d) + \varepsilon m^2\log(m)/\log^{3/2}(1/\delta))$ (follows from Lemma A.4), the DP-ERM algorithm (Algorithm 4) invoked in step 4 has $O(m)$ gradient steps; each of which takes involves $O(k + \log(m)) = O(\varepsilon m\log(m)/\log^{3/2}(1/\delta))$ operations. That is, the total number of operations of this step is $O(\varepsilon m^2\log(m)/\log^{3/2}(1/\delta))$. Finally, the step 5 requires $O(dk) = O(\varepsilon dm\log(m)/\log^{3/2}(1/\delta))$. Thus, the overall running time is $O\left(md\log(\max(d, m)) + \varepsilon m^2\log(m)/\log^{\frac{3}{2}}(1/\delta)\right)$.

$\square$

## C DP Algorithms for Kernel-Based Classification with Margin Guarantees

**Theorem 4.2.** *Let $r > 0$. Let $K : \mathcal{X} \times \mathcal{X} \to \mathbb{R}$ be a shift-invariant, positive definite kernel, where $K(x, x) = r^2$ for all $x \in \mathcal{X}$. For any $\varepsilon > 0$ and $\delta \in (0, 1)$, Algorithm 3 is $(\varepsilon, \delta)$-differentially private. Define $\mathcal{H}_\Lambda \triangleq \{h \in \mathbb{H} : \|w\|_{\mathbb{H}} \le \Lambda\}$, where $\|\cdot\|_{\mathbb{H}}$ is the norm corresponding to the reproducing kernel Hilbert space (RKHS) $\mathbb{H}$ associated with the kernel $K$. Let $\beta \in (0, 1)$. Given an input sample $S$ of $m$ examples drawn i.i.d. from a distribution $\mathcal{D}$ over $\mathcal{X} \times \{\pm 1\}$, Algorithm 3 outputs $h_{w^{\mathrm{Priv}}}^{\widehat{\psi}}$ such that with probability at least $1 - \beta$, we have*

$$R_{\mathcal{D}}(h_{w^{\mathrm{Priv}}}^{\widehat{\psi}}) \le \min_{h \in \mathcal{H}_\Lambda} \widehat{L}_S^\rho(h) + \widetilde{O}\left(\frac{\Lambda r}{\rho \sqrt{\min(1, \varepsilon)\, m}}\right),$$

*where, for any $h \in \mathcal{H}_\Lambda$, $\widehat{L}_S^\rho(h) \triangleq \frac{1}{m} \sum_{i=1}^m \ell^\rho\left(y_i \langle h, \psi(x_i) \rangle_{\mathbb{H}}\right)$, where $\psi$ is the feature map associated with the kernel $K$ and $\langle \cdot, \cdot \rangle_{\mathbb{H}}$ is the inner product associated with the RKHS $\mathbb{H}$.*

A more precise version of the above bound is given as follows:

$$R_{\mathcal{D}}(h_{w^{\mathrm{Priv}}}^{\widehat{\psi}}) \le \min_{h \in \mathcal{H}_\Lambda} \widehat{L}_S^\rho(h) + O\left(\sqrt{\frac{\log(1/\beta)}{m}} + \frac{\Lambda r}{\rho}\left(\frac{1}{\sqrt{m}} + \frac{\sqrt{\log(\frac{m}{\beta})\log(\frac{1}{\beta})}\log^{\frac{3}{4}}(\frac{1}{\delta})}{\sqrt{\varepsilon m}}\right)\right),$$

*Proof.* Let $S_{\widehat{\psi}}$ be as defined in step 4 in Algorithm 3. Note that by Theorem 4.1, we have $\|\widehat{\psi}(x_i)\|_2 = r$ for all $i \in [m]$. Moreover, the output of Algorithm 3 depends only on $S_{\widehat{\psi}}$. Thus, the privacy guarantee follows directly from the privacy guarantee of Algorithm 2 (Theorem 3.2).

Next, we turn to proving the claimed margin bound. First, note that using the margin bound attained by Algorithm 2 (Theorem 3.2), it follows that for any fixed realization of the randomness in $\widehat{\psi}$, with probability $1 - \beta/2$ over the choice of $S \sim \mathcal{D}^m$ and the internal randomness of Algorithm 2, we have

$$R_{\mathcal{D}}(h_{w^{\mathrm{Priv}}}^{\widehat{\psi}}) \le \min_{w \in \mathbb{B}^{2D}(2\Lambda)} \widehat{L}_{S_{\widehat{\psi}}}^\rho(w) + O\left(\sqrt{\frac{\log(1/\beta)}{m}} + \frac{\Lambda r}{\rho}\left(\frac{1}{\sqrt{m}} + \frac{\sqrt{\log(\frac{m}{\beta})\log(\frac{1}{\beta})}\log^{\frac{3}{4}}(\frac{1}{\delta})}{\sqrt{\varepsilon m}}\right)\right). \tag{17}$$

The essence of the proof is to show that with probability $\ge 1 - \beta/2$ over the randomness in $\widehat{\psi}$ (i.e., over the choice of $\omega_1, \ldots, \omega_D$), we have

$$\min_{w \in \mathbb{B}^{2D}(2\Lambda)} \widehat{L}_{S_{\widehat{\psi}}}^\rho(w) \le \min_{h \in \mathcal{H}_\Lambda} \widehat{L}_S^\rho(h) + \frac{2\Lambda r}{\rho \sqrt{m}}. \tag{18}$$

Combining (17) and (18) yields the desired bound.

To prove the bound in (18), we will use the following fact. .

**Fact C.1.** *Let $\mu > 0$. Let $\psi \colon \mathcal{X} \to \mathbb{R}$ denote the feature map associated with the kernel $K$. Let $h_\mu = \mathrm{argmin}_{h \in \mathcal{H}_\Lambda} \left(\widehat{L}_S^\rho(h) + \mu \|h\|_{\mathbb{H}}^2\right)$. Then, $h_\mu = \sum_{i=1}^m \alpha_i \psi(x_i)$ for some $\alpha_i$, $i \in [m]$, that satisfy: $0 \le y_i \alpha_i \le \frac{1}{2m\mu\rho}$.*

This fact simply follows from the dual formulation of the optimization problem for kernel support vector machines (see, e.g., [MRT18, Section 6.3]) . The fact asserts that the minimizer $h_\mu$ of the regularized empirical hinge loss can be expressed as a linear combination of $(\psi(x_i) : i \in [m])$ (such assertion also follows from the representer theorem) where the coefficients of the linear combination (the dual variables) $\alpha = (\alpha_1, \ldots, \alpha_m)$ are bounded; namely, $\|\alpha\|_1 \le \frac{1}{2\mu\rho}$.

Below, we set $\mu = \frac{r}{\Lambda \rho \sqrt{m}}$. Let $\widehat{w} = \sum_{i=1}^m \alpha_i \widehat{\psi}(x_i)$ be a $2D$-dimensional approximation of $h_\mu$. Observe that

$$\widehat{L}_{S_{\widehat{\psi}}}^\rho(\widehat{w}) - \widehat{L}_S^\rho(h_\mu) = \frac{1}{m} \sum_{i=1}^m \left[\ell^\rho(y_i\langle \widehat{w}, \widehat{\psi}(x_i)\rangle) - \ell^\rho(y_i\langle h_\mu, \psi(x_i)\rangle_{\mathbb{H}})\right]$$

$$\le \frac{1}{\rho m} \sum_{i,j\in[m]} |\alpha_j| \cdot |\langle\widehat{\psi}(x_i), \widehat{\psi}(x_j)\rangle - \langle\psi(x_i), \psi(x_j)\rangle_{\mathbb{H}}|$$

$\alpha_i$ where the inequality in the second line follows from the fact that $\ell^\rho$ is $\frac{1}{\rho}$-Lipschitz. Hence, by Theorem 4.1, with probability $\geq 1 - \beta/2$ with respect to the randomness in $\widehat{\psi}$, we have

$$\widehat{L}^\rho_{S_{\widehat{\psi}}}(\widehat{w}) - \widehat{L}^\rho_S(h_\mu) \leq \frac{2r^2}{\rho}\sqrt{\frac{\log(2m/\beta)}{D}}\|\alpha\|_1$$

$$\leq \frac{r^2}{\rho^2\mu}\sqrt{\frac{\log(2m/\beta)}{D}}$$

$$\leq \frac{\Lambda r}{\rho\sqrt{m}},$$

where the second inequality follows from the fact that $\|\alpha\|_1 \leq \frac{1}{2\mu\rho}$, which follows from Fact C.1, and the third inequality follows from the setting of $D$ in step 2 in Algorithm 3 and the setting of $\mu = \frac{r}{\Lambda\rho\sqrt{m}}$. Moreover, we note that $\widehat{w} \in \mathbb{B}^{2D}(2\Lambda)$. Indeed, conditioned on the same event above (the kernel matrix is well approximated via $\widehat{\psi}$), observe that

$$\|\widehat{w}\|_2^2 = \sum_{i,j} \alpha_i \alpha_j \langle \widehat{\psi}(x_i), \widehat{\psi}(x_j) \rangle$$

$$\leq \sum_{i,j} \alpha_i \alpha_j \langle \psi(x_i), \psi(x_j) \rangle_{\mathbb{H}} + 2r^2\sqrt{\frac{\log(2m/\beta)}{D}}\|\alpha\|_1^2$$

$$\leq \|h\|_{\mathbb{H}}^2 + \frac{\Lambda^2}{2} \leq \frac{3}{2}\Lambda^2.$$

Thus, we have $\|\widehat{w}\|_2 < 2\Lambda$. Hence, we can assert that with probability $\geq 1 - \beta/2$ over the randomness in $\widehat{\psi}$, we have

$$\min_{w \in \mathbb{B}^{2D}(2\Lambda)} \widehat{L}^\rho_{S_{\widehat{\psi}}}(w) \leq \widehat{L}^\rho_{S_{\widehat{\psi}}}(\widehat{w}) \leq \widehat{L}(h_\mu; S) + \frac{\Lambda r}{\rho\sqrt{m}}.$$

Finally, note that $\widehat{L}^\rho_S(h_\mu) \leq \min_{h \in \mathcal{H}_\Lambda} \widehat{L}^\rho_S(h) + \mu\Lambda^2 = \min_{h \in \mathcal{H}_\Lambda} \widehat{L}^\rho_S(h) + \frac{\Lambda r}{\rho\sqrt{m}}$. Hence, we arrive at the claimed bound (18), and thus, the proof is complete. $\qquad\square$

## D  DP Algorithms for Learning Neural Networks with Margin Guarantees

**Theorem 5.1.** *Let $\varepsilon > 0, \beta \in (0,1)$, and $\rho > 0$. Then, there is an $\varepsilon$-DP algorithm which returns an $L$-layer network $h^{\mathsf{Priv}}$ with $N$ neurons per layer that with probability at least $1 - \beta$ over the draw of a sample $S \sim \mathcal{D}^m$ and the internal randomness of the algorithm admits the following guarantee:*

$$R_{\mathcal{D}}(h^{\mathsf{Priv}}) \leq \min_{h \in \mathcal{H}_{\mathsf{NN}\Lambda}} \widehat{R}^\rho_S(h) + O\left(\frac{r(2\eta\Lambda)^L\sqrt{N\theta}}{\rho\sqrt{m}} + \frac{r^2(2\eta\Lambda)^{2L}N\theta}{\rho^2\varepsilon m}\right),$$

*where $\theta = \log(Lm/\beta)\log(r(\eta\Lambda)^L/\rho)$.*

*Proof.* First, note that our construction is indeed $\varepsilon$-DP by the properties of the exponential mechanism. Thus, we now turn to the proof of the margin bound. Our proof relies on the following properties of the JL-transform.

**Lemma D.1** (Follows from Theorem 109 in [Nel10]). *Let $p, N, m, k \in \mathbb{N}$. Let $W \in \mathbb{R}^{p \times N}$. Let $z_1, \ldots, z_m \in \mathbb{R}^p$. Let $\Phi$ be a random $k \times p$ matrix with entries drawn i.i.d. uniformly from $\{\pm\frac{1}{\sqrt{k}}\}$. Let $\beta \in (0,1)$. There is a constant $c > 0$ such that the following inequalities hold simultaneously with probability at least $1 - \beta$:*

$$\|\Phi W\|_F^2 \leq \|W\|_F^2\left(1 + c\sqrt{\frac{\log(m/\beta)}{k}}\right),$$

$$\forall i \in [m]: \|W^\top\Phi^\top\Phi z_i - W^\top z_i\|_2 \leq c\|W\|_F\|z_i\|_2\sqrt{\frac{\log(m/\beta)}{k}}$$

Consider the algorithmic construction described earlier. Let $h_* \in \underset{h \in \mathcal{H}_{\mathsf{NN}}}{\operatorname{argmin}} \widehat{R}_S^\rho(h)$. Let $W_1^*, \ldots, W_L^*$ denote the weight matrices of $h_*$. Let $h_*^\Phi \in \mathcal{H}_{\mathsf{NN}}^\Phi$ be the network specified by the matrices $\widetilde{W}_1 \triangleq \Phi_0 W_1^*, \ldots, \widetilde{W}_L \triangleq \Phi_{L-1} W_L^*$. That is, the weight matrices of $h_*^\Phi$ are given by $\Phi_0^\top \widetilde{W}_1 = \Phi_0^\top \Phi_0 W_1^*, \ldots,$ $\Phi_{L-1}^\top \widetilde{W}_L = \Phi_{L-1}^\top \Phi_{L-1} W_L^*$.

We make the following four claims. Combining those claims together with the union bound immediately yields the margin bound of the theorem. We first state those claims and then prove them.

**Claim D.2.** *There is a setting $k = O\left(\frac{r^2 (2\eta\Lambda)^{2L} \log(Lm/\beta)}{\rho^2}\right)$ such that with probability $1 - \beta/4$ over the choice of $\Phi_0, \ldots, \Phi_{L-1}$, we have $h_*^\Phi \in \mathcal{H}_{\mathsf{NN}^{2\Lambda}}^\Phi$ and for all $i \in [m]$*

$$|h_*(x_i) - h_*^\Phi(x_i)| = O\left(r(2\eta\Lambda)^L \sqrt{\frac{\log(Lm/\beta)}{k}}\right).$$

*Consequently, with probability $1 - \beta/4$,*

$$\widehat{R}_S^{0.5\rho}(h_*^\Phi) \leq \widehat{R}_S^\rho(h_*).$$

**Claim D.3.** *Let $\widehat{h}^\Phi \in \underset{h \in \widehat{\mathcal{H}}_{\mathsf{NN}^{2\Lambda}}^\Phi}{\operatorname{argmin}} \widehat{R}_S(h)$. There exists a setting $k = O\left(\frac{r^2 (2\eta\Lambda)^{2L} \log(Lm/\beta)}{\rho^2}\right)$ for the embedding parameter such that with probability $1 - \beta/4$*

$$\widehat{R}_S(\widehat{h}^\Phi) \leq \widehat{R}_S^{0.5\rho}(h_*^\Phi).$$

**Claim D.4.** *Let $\widehat{h}^\Phi \in \underset{h \in \widehat{\mathcal{H}}_{\mathsf{NN}^{2\Lambda}}^\Phi}{\operatorname{argmin}} \widehat{R}_S(h)$. Let $k = O\left(\frac{r^2 (2\eta\Lambda)^{2L} \log(Lm/\beta)}{\rho^2}\right)$. With probability $1 - \beta/4$ over the randomness of the exponential mechanism, we have*

$$\widehat{R}_S(h^{\mathsf{Priv}}) \leq \widehat{R}_S(\widehat{h}^\Phi) + O\left(\frac{r^2 (2\eta\Lambda)^{2L} N \log(Lm/\beta) \log(r(4\eta\Lambda)^L/\rho)}{\rho^2 \varepsilon m}\right).$$

**Claim D.5.** *Let $k = O\left(\frac{r^2 (2\eta\Lambda)^{2L} \log(Lm/\beta)}{\rho^2}\right)$. With probability $1 - \beta/4$ over the choice of $S \sim \mathcal{D}^m$, we have*

$$R_\mathcal{D}(h^{\mathsf{Priv}}) \leq \widehat{R}_S(h^{\mathsf{Priv}}) + O\left(\frac{r(2\eta\Lambda)^L \sqrt{N \log(Lm/\beta)} \log(r(\eta\Lambda)^L/\rho)}{\rho \sqrt{m}}\right).$$

Recall that $\widehat{\mathcal{H}}_{\mathsf{NN}^{2\Lambda}}^\Phi$ is a finite approximation of $\mathcal{H}_{\mathsf{NN}^{2\Lambda}}^\Phi$ constructed via a $\gamma$-cover $\mathcal{C}$ for $\mathbb{B}^{k \times N}(2\Lambda) \times \ldots \times \mathbb{B}^k(2\Lambda)$, where we choose $\gamma = \frac{\rho}{10r(4\eta\Lambda)^{L-1}}$. In particular, for any $W = (W_1, \ldots, W_L), W' = (W_1', \ldots, W_L') \in \mathcal{C}$, we have $\|W - W'\|_F = \sqrt{\sum_{j=1}^L \|W_j - W_j'\|_F^2} \leq \gamma$. Given that $\mathcal{C}$ is a $\gamma$-cover, we have $|\widehat{\mathcal{H}}_{\mathsf{NN}^{2\Lambda}}^\Phi| = |\mathcal{C}| = O\left(\left(\frac{\sqrt{L}\Lambda}{\gamma}\right)^{k \times N}\right)$. Namely, $\log(|\widehat{\mathcal{H}}_{\mathsf{NN}^{2\Lambda}}^\Phi|) = O\left(kN \log(\frac{r(\eta\Lambda)^L}{\rho})\right)$. Given this, together with the setting of $k$ in Claim D.5 and the fact that $h^{\mathsf{Priv}}$ is in $\widehat{\mathcal{H}}_{\mathsf{NN}^{2\Lambda}}^\Phi$, note that Claim D.5 follows from a straightforward uniform convergence bound for the hypotheses in $\widehat{\mathcal{H}}_{\mathsf{NN}^{2\Lambda}}^\Phi$. Note also that the proof of Claim D.4 follows directly from the standard accuracy guarantee of the exponential mechanism when instantiated on $\widehat{\mathcal{H}}_{\mathsf{NN}^{2\Lambda}}^\Phi$. In particular, since the score function is $-\widehat{R}_S(\cdot)$, with probability at least $1 - \beta/4$, the excess empirical loss of $h^{\mathsf{Priv}}$ is bounded by $O\left(\frac{\log(|\mathcal{C}|/\beta)}{\varepsilon m}\right)$, which yields the bound claimed in Claim D.4 given the bound on $|\mathcal{C}|$ above and the setting of $k$.

We now turn to the proofs of Claims D.2 and D.3. We start with the proof of Claim D.2.

For each $i \in [m]$ and each $j \in [L]$, let $v_{i,j} \in \mathbb{R}^N$ denote the output of the $j$-th layer of $h_*$ on input $x_i$ prior to activation (i.e., $v_{i,j}$ is the input to the neurons of layer $j + 1$ when the input to the network $h_*$ is the $i$-th feature vector $x_i$ in the dataset $S$). Analogously, for each $i \in [m]$ and each $j \in [L]$, let $v_{i,j}^\Phi$ denote the output of the $j$-th layer of $h_*^\Phi$ on input $x_i$ prior to activation. Also, let $u_{i,j} \triangleq \psi(v_{i,j}) - \psi(v_{i,j}^\Phi), i \in [m], j \in [L]$.

As a direct corollary of Lemma D.1, by applying the union bound over the choice of $\Phi_0, \ldots, \Phi_{L-1}$, there is a constant $\hat{c} > 0$ such that with probability $1 - \beta/4$ over the choice of $\Phi_0, \ldots, \Phi_{L-1}$, for all $i \in [m], j \in [L]$, we have

$$\|\Phi_{j-1} W_j^*\|_F^2 \leq \Lambda^2 \left(1 + \hat{c}\sqrt{\frac{\log(Lm/\beta)}{k}}\right), \tag{19}$$

$$\|(W_j^*)^\top \Phi_{j-1}^\top \Phi_{j-1} \psi(v_{i,j-1}) - (W_j^*)^\top \psi(v_{i,j-1})\|_2 \leq \hat{c}\Lambda \|\psi(v_{i,j-1})\|_2 \sqrt{\frac{\log(Lm/\beta)}{k}}, \ \ j \neq 1, \tag{20}$$

$$\|(W_j^*)^\top \Phi_{j-1}^\top \Phi_{j-1} u_{i,j-1} - (W_j^*)^\top u_{i,j-1}\|_2 \leq \hat{c}\Lambda \|u_{i,j-1}\|_2 \sqrt{\frac{\log(Lm/\beta)}{k}}, \ \ j \neq 1, \tag{21}$$

$$\|(W_1^*)^\top \Phi_0^\top \Phi_0 x_i - (W_1^*)^\top x_i\|_2 \leq \hat{c}\Lambda r \sqrt{\frac{\log(Lm/\beta)}{k}} \tag{22}$$

We now condition on the event where all the above inequalities are satisfied for the remainder of the proof. Below, we let $\tau = \hat{c}\sqrt{\frac{\log(Lm/\beta)}{k}} < 1$. First, from (19), there is a setting $k$ as indicated in the statement of the claim, where $\|\Phi_{j-1} W_j^*\|_F < 2\Lambda$. Thus, $h_*^\Phi \in \mathcal{H}_{\mathrm{NN}^{2\Lambda}}^\Phi$.

Now, fix any $i \in [m]$. Define $\Gamma_j \triangleq \|(W_j^*)^\top \psi(v_{i,j-1}) - (W_j^*)^\top \Phi_{j-1}^\top \Phi_{j-1} \psi(v_{i,j-1}^\Phi)\|_2$ for $j \in [L]$. Observe that

$$|h_*(x_i) - h_*^\Phi(x_i)| = \Gamma_L$$
$$= |(W_L^*)^\top \psi(v_{i,L-1}) - (W_L^*)^\top \Phi_{L-1}^\top \Phi_{L-1} \psi(v_{i,L-1}^\Phi)|$$
$$\leq |(W_L^*)^\top \psi(v_{i,L-1}) - (W_L^*)^\top \Phi_{L-1}^\top \Phi_{L-1} \psi(v_{i,L-1})|$$
$$\quad + |(W_L^*)^\top \Phi_{L-1}^\top \Phi_{L-1} \psi(v_{i,L-1}) - (W_L^*)^\top \Phi_{L-1}^\top \Phi_{L-1} \psi(v_{i,L-1}^\Phi)|$$
$$\leq \tau \Lambda \|\psi(v_{i,L-1})\|_2 + |(W_L^*)^\top \Phi_{L-1}^\top \Phi_{L-1} \left(\psi(v_{i,L-1}) - \psi(v_{i,L-1}^\Phi)\right)| \quad \left(\text{follows from (20) and the fact } W_L^* \in \mathbb{B}^N(\Lambda)\right)$$
$$= \tau \Lambda \|\psi(v_{i,L-1})\|_2 + |(W_L^*)^\top \Phi_{L-1}^\top \Phi_{L-1} u_{i,L-1}| \quad \left(\text{by definition of } u_{i,L-1} \text{ given above}\right)$$
$$\leq \tau \Lambda \|\psi(v_{i,L-1})\|_2 + (1 + \tau)\Lambda \|u_{i,L-1}\|_2 \quad \left(\text{follows from (21)}\right)$$
$$\leq \tau \Lambda \|\psi(v_{i,L-1})\|_2 + 2\Lambda \|\psi(v_{i,L-1}) - \psi(v_{i,L-1}^\Phi)\|_2$$
$$\leq \tau \eta \Lambda \|v_{i,L-1}\|_2 + 2\eta \Lambda \|v_{i,L-1} - v_{i,L-1}^\Phi\|_2 \quad \left(\text{since } \psi \text{ is } \eta\text{-Lipschitz and } \psi(0) = 0\right)$$
$$= \tau \eta \Lambda \|v_{i,L-1}\|_2 + 2\eta \Lambda \|(W_{L-1}^*)^\top \psi(v_{i,L-2}) - (W_{L-1}^*)^\top \Phi_{L-2}^\top \Phi_{L-2} \psi(v_{i,L-2}^\Phi)\|_2$$
$$= \tau \eta \Lambda \|v_{i,L-1}\|_2 + 2\eta \Lambda \Gamma_{L-1}$$

Hence, we obtain $\Gamma_L \leq \tau \eta \Lambda \|v_{i,L-1}\|_2 + 2\eta \Lambda \Gamma_{L-1}$. Before we solve this recurrence, we first unravel the term $\|v_{i,L-1}\|_2$. Note that

$$\|v_{i,L-1}\|_2 = \|(W_{L-1}^*)^\top \psi(v_{i,L-2})\|_2$$
$$\leq \|W_{L-1}^*\|_F \cdot \|\psi(v_{i,L-2})\|_2$$
$$\leq \eta \Lambda \|v_{i,L-2}\|_2$$

Proceeding recursively, we obtain

$$\|v_{i,L-1}\|_2 \leq \eta^{L-2} \Lambda^{L-1} \|x_i\|_2 \leq r \eta^{L-2} \Lambda^{L-1}.$$

Plugging this in the recurrence for $\Gamma_L$ above yields

$$\Gamma_L \leq \tau r \eta^{L-1} \Lambda^L + 2\eta \Lambda \Gamma_{L-1}.$$

Unraveling this recursion (and using (22) in the last step of the recursion) yields

$$|h_*(x_i) - h_*^\Phi(x_i)| = \Gamma_L \leq r(2\eta\Lambda)^{L-1} \Lambda \tau = \hat{c}\sqrt{\frac{\log(Lm/\beta)}{k}} r(2\eta\Lambda)^{L-1} \Lambda.$$

Note that choosing $k = \frac{10\hat{c}^2 r^2 (2\eta\Lambda)^{2(L-1)} \Lambda^2 \log(Lm/\beta)}{\rho^2}$ guarantees $|h_*(x_i) - h_*^\Phi(x_i)| < \frac{\rho}{2}$ for all $i \in [m]$. Hence, as in the argument of the proof of Theorem 3.1, this implies that for all $i \in [m]$, $y_i h_*(x_i) > \rho \Rightarrow y_i h_*^\Phi(x_i) > \frac{\rho}{2}$. Thus, $\widehat{R}_S^{0.5\rho}(h_*^\Phi) \leq \widehat{R}_S^\rho(h_*)$. This concludes the proof of Claim D.2.

Finally, we prove Claim D.3.

As shown in Claim D.2, we have $h_*^\Phi \in \mathcal{H}_{\mathsf{NN}^{2\Lambda}}^\Phi$. Since $\widehat{\mathcal{H}}_{\mathsf{NN}^{2\Lambda}}^\Phi$ is a $\gamma$-cover of $\mathcal{H}_{\mathsf{NN}^{2\Lambda}}^\Phi$, there exists $\tilde{h} \in \widehat{\mathcal{H}}_{\mathsf{NN}^{2\Lambda}}^\Phi$ that "approximates" $h_*^\Phi$. Namely, there is $\tilde{h} \in \widehat{\mathcal{H}}_{\mathsf{NN}^{2\Lambda}}^\Phi$ defined by matrices $(\widetilde{W}_1, \ldots, \widetilde{W}_L) \in \mathcal{C}$ such that

$$\sum_{j=1}^{L} \|\widetilde{W}_j - \Phi_{j-1} W_j^*\|_F^2 \le \gamma^2,$$

where, as defined before, $(\Phi_0 W_1^*, \ldots, \Phi_{L-1} W_L^*)$ are the matrices defining $h_*^\Phi$. We choose $\gamma = \frac{\rho}{10 r (4\eta\Lambda)^{L-1}}$.

To simplify notation, we will denote

$$W_j^{\Phi,*} \triangleq \Phi_{j-1} W_j^*, \ \forall \ j \in [L].$$

As before, for each $i \in [m], j \in [L]$, we let $v_{i,j}^\Phi \in \mathbb{R}^N$ denote the output of the $j$-th layer of $h_*^\Phi$ on input $x_i$ prior to activation, and let $\tilde{v}_{i,j}$ denote the output of the $j$-th layer of $\tilde{h}$ on input $x_i$ prior to activation.

Again, as a corollary of Lemma D.1 (by applying the union bound over the choice of $\Phi_0, \ldots, \Phi_{L-1}$), there is a constant $\hat{c} > 0$ such that with probability $1 - \beta/4$ over the choice of $\Phi_0, \ldots, \Phi_{L-1}$, for all $i \in [m], j \in [L]$, we have

$$\|\Phi_{j-1} \psi(\tilde{v}_{i,j-1})\|_2^2 \le \|\psi(\tilde{v}_{i,j-1})\|_2^2 \left(1 + \hat{c}\sqrt{\frac{\log(Lm/\beta)}{k}}\right), \ j \ne 1 \tag{23}$$

$$\|\Phi_{j-1} \left(\psi(\tilde{v}_{i,j-1}) - \psi(v_{i,j-1}^\Phi)\right)\|_2^2 \le \|\psi(\tilde{v}_{i,j-1}) - \psi(v_{i,j-1}^\Phi)\|_2^2 \left(1 + \hat{c}\sqrt{\frac{\log(Lm/\beta)}{k}}\right), \ j \ne 1 \tag{24}$$

$$\|\Phi_0 x_i\|_2^2 \le r^2 \left(1 + \hat{c}\sqrt{\frac{\log(Lm/\beta)}{k}}\right) \tag{25}$$

We will condition on the event above for the remainder of the proof. Note that for the setting of $k$ as in Claim D.2, we have $\left(1 + \hat{c}\sqrt{\frac{\log(Lm/\beta)}{k}}\right) < 2$.

For each $j \in [L]$, define

$$\Delta_j \triangleq \|\widetilde{W}_j^\top \Phi_{j-1} \psi(\tilde{v}_{i,j-1}) - W_j^{\Phi,*} \Phi_{j-1} \psi(v_{i,j-1}^\Phi)\|_2.$$

Fix any $i \in [m]$. Observe

$$|\tilde{h}(x_i) - h_*^\Phi(x_i)| = \Delta_L$$
$$\le |\widetilde{W}_L^\top \Phi_{L-1} \psi(\tilde{v}_{i,L-1}) - (W_L^{\Phi,*})^\top \Phi_{L-1} \psi(\tilde{v}_{i,L-1})|$$
$$\quad + |(W_L^{\Phi,*})^\top \Phi_{L-1} \psi(\tilde{v}_{i,L-1}) - \widetilde{(W_L^{\Phi,*})}^\top \Phi_{L-1} \psi(v_{i,L-1}^\Phi)|$$
$$\le \|\widetilde{W}_L - W_L^{\Phi,*}\|_F \|\Phi_{L-1} \psi(\tilde{v}_{i,L-1})\|_2 + \|W_L^{\Phi,*}\|_F \|\Phi_{L-1}\left(\psi(\tilde{v}_{i,L-1}) - \psi(v_{i,L-1}^\Phi)\right)\|_2$$
$$\le \gamma \|\Phi_{L-1} \psi(\tilde{v}_{i,L-1})\|_2 + 2\Lambda \|\Phi_{L-1}\left(\psi(\tilde{v}_{i,L-1}) - \psi(v_{i,L-1}^\Phi)\right)\|_2 \quad (\mathcal{C} \text{ is } \gamma\text{-cover and } W_L^{\Phi,*} \in \mathbb{B}^{k \times N}(2\Lambda))$$
$$\le \sqrt{2}\gamma \|\psi(\tilde{v}_{i,L-1})\|_2 + 2\sqrt{2}\Lambda \|\psi(\tilde{v}_{i,L-1}) - \psi(v_{i,L-1}^\Phi)\|_2 \quad (\text{follows from (23)-(24)})$$
$$\le \sqrt{2}\gamma\eta \|\tilde{v}_{i,L-1}\|_2 + 2\sqrt{2}\eta\Lambda \|\tilde{v}_{i,L-1} - v_{i,L-1}^\Phi\|_2 \quad (\psi \text{ is } \eta\text{-Lipschitz and } \psi(0) = 0)$$
$$\le \sqrt{2}\gamma\eta \|\widetilde{W}_{L-1}^\top \Phi_{L-2} \psi(\tilde{v}_{i,L-2})\|_2 + 2\sqrt{2}\eta\Lambda \|\widetilde{W}_{L-1}^\top \Phi_{L-2} \psi(\tilde{v}_{i,L-2}) - W_{L-1}^{\Phi,*} \Phi_{L-2} \psi(v_{i,L-2}^\Phi)\|_2$$
$$= \sqrt{2}\gamma\eta \|\widetilde{W}_{L-1}^\top \Phi_{L-2} \psi(\tilde{v}_{i,L-2})\|_2 + 2\sqrt{2}\eta\Lambda \Delta_{L-1}$$

Hence, we arrive at a recursive bound

$$\Delta_L \le \sqrt{2}\gamma\eta \|\widetilde{W}_{L-1}^\top \Phi_{L-2} \psi(\tilde{v}_{i,L-2})\|_2 + 2\sqrt{2}\eta\Lambda \Delta_{L-1}.$$

Before proceeding, we first unravel the term $\|\widetilde{W}_{L-1}^\top \Phi_{L-2}\psi(\tilde{v}_{i,L-2})\|_2$. Let's denote this term as $B_{L-1}$. Observe that

$$
\begin{aligned}
B_{L-1} &= \|\widetilde{W}_{L-1}^\top \Phi_{L-2}\psi(\tilde{v}_{i,L-2})\|_2 \\
&\leq \|\widetilde{W}_{L-1}\|_F \|\Phi_{L-2}\psi(\tilde{v}_{i,L-2})\|_2 \\
&\leq 2\sqrt{2}\Lambda \|\psi(\tilde{v}_{i,L-2})\|_2 \\
&\leq 2\sqrt{2}\eta\Lambda \|\tilde{v}_{i,L-2}\|_2 \\
&= 2\sqrt{2}\eta\Lambda \|\widetilde{W}_{L-2}^\top \Phi_{L-3}\psi(\tilde{v}_{i,L-3})\|_2 \\
&= 2\sqrt{2}\eta\Lambda B_{L-2}
\end{aligned}
$$

Thus, continuing recursively, we get $B_{L-1} \leq r\eta^{L-2}(2\sqrt{2}\Lambda)^{L-1}$ (where in the last step of the recursion, we use (25)). Plugging this back in the recursive bound for $\Delta_L$, we get

$$
\Delta_L \leq \sqrt{2}\gamma r (2\sqrt{2}\eta\Lambda)^{L-1} + 2\sqrt{2}\,\eta\,\Lambda\,\Delta_{L-1}
$$

Unraveling this recurrence yields

$$
\begin{aligned}
\Delta_L &\leq \sqrt{2}\gamma r L (2\sqrt{2}\eta\Lambda)^{L-1} \\
&\leq 2\sqrt{2}\gamma r (4\eta\Lambda)^{L-1}
\end{aligned}
$$

Thus, by the choice of $\gamma$, we have $|\tilde{h}(x_i) - h_*^\Phi(x_i)| < \frac{\rho}{2}$ for all $i \in [m]$. Hence, as before, we have $\left(y_i h_*^\Phi(x_i) > \rho/2\right) \Rightarrow \left(y_i \tilde{h}(x_i) > 0\right)$ for all $i \in [m]$, which implies that

$$
\widehat{R}_S(\tilde{h}) \leq \widehat{R}_S^{0.5\rho}(h_*^\Phi).
$$

Since $\widehat{h}^\Phi \in \underset{h \in \widehat{\mathcal{H}}_{\mathsf{NN}^{2\Lambda}}^\Phi}{\operatorname{argmin}} \widehat{R}_S(h)$, then we have $\widehat{R}_S(\widehat{h}^\Phi) \leq R_S(\tilde{h})$. Therefore, we can write

$$
\widehat{R}_S(\widehat{h}^\Phi) \leq \widehat{R}_S^{0.5\rho}(h_*^\Phi).
$$

This concludes the proof of Claim D.3 and completes the proof of Theorem 5.1. $\qquad\square$

# E  Label-Private Algorithms with Margin Guarantees

In many tasks, the features are public information and only the labels are sensitive and need to be protected. Several recent publications have suggested to train learning models with differential privacy for labels for these tasks, while treating features as public information [GGK+21, EMSV21]. This motivates the following definition of label differential privacy.

**Definition E.1** (Label differential privacy). *Let $\varepsilon, \delta \geq 0$. Let $\mathcal{A}\colon (\mathcal{X} \times \mathcal{Y})^m \to \mathcal{H}$ be a (potentially randomized) mechanism. We say that $\mathcal{A}$ is $(\varepsilon, \delta)$-label-DP if for any measurable subset $O \subset \mathcal{H}$ and all $S, S' \in (\mathcal{X} \times \mathcal{Y})^m$ that differ in one label of one sample, the following inequality holds:*

$$
\mathbb{P}(\mathcal{A}(S) \in O) \leq e^\varepsilon\,\mathbb{P}(\mathcal{A}(S') \in O) + \delta. \tag{26}
$$

[GGK+21] gave an algorithm for deep learning with label differential privacy in the local differential privacy model. [YSMN21] proposed and evaluated algorithms for label differential privacy in conjunction with secure multiparty computation. [EMSV21] presented a clustering-based algorithm for label differential privacy. There are several other works which show pitfalls on label differential privacy [BFSV+21a, BFSV+21b].

Here, we design a simple algorithm for label differential privacy, which we show benefits from margin guarantees for any hypotheses class with finite fat-shattering dimension, including the class of linear classifiers, neural networks, and ensembles [BST99].

We first introduce some definitions needed to describe our algorithm. Fix $\rho > 0$. Define the $\rho$-truncation function $\beta_\rho\colon \mathbb{R} \to [-\rho, +\rho]$ by $\beta_\rho(u) = \max\{u, -\rho\}1_{u\leq 0} + \min\{u, +\rho\}1_{u\geq 0}$, for all $u \in \mathbb{R}$. For any $h \in \mathcal{H}$, we denote by $h_\rho$ the $\rho$-truncation of $h$, $h_\rho = \beta_\rho(h)$, and define $\mathcal{H}_\rho = \{h_\rho\colon h \in \mathcal{H}\}$. For any family of functions $\mathcal{F}$, we also denote by $\mathcal{N}_\infty(\mathcal{F}, \varepsilon, x_1^m)$ the empirical covering number of $\mathcal{F}$ over the sample $(x_1, \ldots, x_m)$ and by $\mathcal{C}(\mathcal{F}, \varepsilon, x_1^m)$ a minimum empirical cover. With these definitions, the algorithm is given in Algorithm 5. The algorithm uses an exponential mechanism over a cover of truncated hypotheses sets.

---

**Algorithm 5** $\mathcal{A}_{\mathsf{LabMarg}}$: Private learning algorithm under label-privacy

---

**Require:** Dataset $S = ((x_1, y_1), \ldots, (x_m, y_m)) \in (B^d \times \{\pm 1\})^m$; privacy parameter $\varepsilon > 0$; margin parameter $\rho$.

1: Compute the $\rho/2$ minimal cover $\widehat{\mathcal{H}}_\rho = \mathcal{C}(\mathcal{H}_\rho, \rho/2, x_1^m)$.
2: Run the Exponential mechanism with privacy parameter $\varepsilon$, sensitivity $1/m$, and score function $-\widehat{R}_S^{\rho/2}(h)$, $h \in \widehat{\mathcal{H}}_\rho$ to select $h^{\mathsf{priv}} \in \widehat{\mathcal{H}}_\rho$.
3: **return** $h^{\mathsf{priv}}$.

---

**Theorem E.1.** *Algorithm 5 is $\varepsilon$-label-DP. Let $\mathcal{D}$ be a distribution on $\mathcal{X} \times \mathcal{Y}$ and suppose $S \sim \mathcal{D}^m$. Let $c = 17$ and $d = \mathrm{fat}_{\frac{\rho}{32}}(\mathcal{H})$ and $M = 1 + d \log_2(2c^2m) \log_2 \frac{2cem}{d} + \log \frac{2}{\beta}$. For any $\beta \in (0, 1)$, with probability at least $1 - \beta$, the output $w^{\mathsf{Priv}}$ satisfies:*

$$R_\mathcal{D}(h^{\mathsf{priv}}) \le \min_{h \in \mathcal{H}}\left( \widehat{R}_S^\rho(h) + 2\sqrt{\min_{h \in \mathcal{H}} \widehat{R}_S^\rho(h)}\sqrt{\frac{M}{m}} \right) + \frac{2M}{m} + \frac{64M \log\left(\frac{2}{\beta}\right)}{\varepsilon m}.$$

Before we prove the above theorem, we first want to remark that while our algorithm is computationally inefficient, it admits strong theoretical guarantees. First, it is an $(\varepsilon, 0)$ pure label-differential privacy guarantee. Second, it is dimension-independent. Furthermore, our algorithm benefits from a relative deviation margin bound that smoothly interpolates between the realizable case of $R_S^\rho(w) = 0$ and the case of $R_S^\rho(w) > 0$. As a corollary, note that up to constants one can always get privacy for $\varepsilon > 1$ for free. Finally, observe that this bound holds not only for linear classes, but also for any hypothesis set with favorable $\rho$-fat-shattering dimension. In particular, we can use known upper bounds for the $\rho$-fat-shattering dimension of feed-forward neural networks [BST99] to derive label-privacy guarantees for training neural networks.

We now prove Theorem E.1.

*Proof.* The $\varepsilon$-differential privacy guarantee follows directly from the properties of the exponential mechanism. In particular, given the finite class $\widehat{\mathcal{H}}_\rho$ and the score function $-\widehat{R}_S^{\rho/2}(h)$, $h \in \widehat{\mathcal{H}}_\rho$, the algorithm becomes an instantiation of the exponential mechanism [MT07].

We focus on proving the utility guarantee in the rest of the proof. If $m < \frac{64M \log(2/\beta)}{\varepsilon}$, then the bound follows trivially. Hence in the rest of the paper, we focus on the regime $m \ge \frac{64M \log(2/\beta)}{\varepsilon}$. By definition of $\widehat{\mathcal{H}}_\rho$, for any $h \in \mathcal{H}$ there exists $g \in \widehat{\mathcal{H}}_\rho$ such that for any $x \in x_1^m$,

$$|g(x) - h(x)| \le \frac{\rho}{2}.$$

Thus, for any $y \in \{-1, +1\}$ and $x \in x_1^m$, we have $|yg(x) - yh(x)| \le \rho/2$, which implies:

$$\mathbb{1}_{yg(x) \le \rho/2} \le \mathbb{1}_{yh(x) \le \rho}.$$

Let $h_S^* \in \mathrm{argmin}_{h \in \mathcal{H}} \widehat{R}_S^\rho(h)$. By the construction of $\widehat{\mathcal{H}}_\rho$ and the above argument,

$$\min_{h \in \widehat{\mathcal{H}}_\rho} \widehat{R}_S^{\rho/2}(h) \le \widehat{R}_S^\rho(h_S^*). \tag{27}$$

We now bound the size $\widehat{\mathcal{H}}_\rho$.

$$|\widehat{\mathcal{H}}_\rho| = \mathcal{N}_\infty(\mathcal{H}_\rho, \rho/2, x_1^m).$$

By [Bar98, Proof of theorem 2], we have

$$\log \max_{x_1^m}[\mathcal{N}_\infty(\mathcal{H}_\rho, \tfrac{\rho}{2}, x_1^m)] \le 1 + d' \log_2(2c^2m) \log_2 \frac{2cem}{d'},$$

where $d' = \mathrm{fat}_{\frac{\rho}{32}}(\mathcal{H}_\rho) \le \mathrm{fat}_{\frac{\rho}{32}}(\mathcal{H}) = d$ and $c = 17$. Given the bound on the sample size in the theorem statement and the properties of the exponential mechanism [MT07], value of $m$, with probability at

---

**Algorithm 6** $\mathcal{A}_{\mathsf{PrivMrg}}$: Algorithm to select confidence margin

---

**Require:** Dataset $S = ((x_1, y_1), \ldots, (x_m, y_m)) \in (\mathcal{X} \times \{\pm 1\})^m$; algorithm $\mathcal{A}$; bound $F(\rho, g_\rho(S))$;
$\;\;\;\;$ $h_{\max}$ an upper bound on $\rho$; privacy parameters $\varepsilon > 0, \delta \geq 0$; and confidence parameter $\beta > 0$.
1: Let $\mathcal{V} \triangleq \left\{ \rho_j \triangleq 2^{-j} \, h_{\max} : j \in [J] \right\}$, where $J = \frac{1}{2} \log(m)$.
2: Run the generalized exponential mechanism [RS16, Algorithm 1] over $\mathcal{V}$ with privacy parameter
$\;\;\;\;$ $\varepsilon$ and score function $-F(\rho_j; g_{\rho_j}(S))$ for $\rho_j \in \mathcal{V}$, to select $\rho^* \in \mathcal{V}$.
3: Run $\mathcal{A}$ on the dataset $S$ with margin parameter $\rho^*$ and privacy parameters $(\varepsilon, \delta)$, confidence
$\;\;\;\;$ parameter $\beta$, and return its output $w^{\mathsf{Priv}}$.

---

least $1 - \beta/2$,

$$\widehat{R}_S^{\rho/2}(h^{\mathsf{priv}}) \leq \min_{h \in \widehat{\mathcal{H}}_\rho} \widehat{R}_S^\rho(h) + \frac{32M \log(2/\beta)}{\varepsilon m}$$

$$\leq \widehat{R}_S^\rho(h_S^*) + \frac{32M \log(2/\beta)}{\varepsilon m}. \tag{28}$$

By Lemma A.2, with probability at least $1 - \beta/2$,

$$R_{\mathcal{D}}(h^{\mathsf{priv}}) \leq \widehat{R}_S^{\rho/2}(h^{\mathsf{priv}}) + 2\sqrt{\widehat{R}_S^{\rho/2}(h) \frac{M}{m}} + \frac{M}{m}, \tag{29}$$

where $M = 1 + d \log_2(2c^2 m) \log_2 \frac{2cem}{d} + \log \frac{2}{\beta}$, $c = 17$, and $d = \mathrm{fat}_{\frac{\rho}{32}}(\mathcal{H})$. Combining (28) and (29)
yields

$$R_{\mathcal{D}}(h^{\mathsf{priv}}) \leq \widehat{R}_S^\rho(h_S^*) + 2\sqrt{\widehat{R}_S^\rho(h_S^*) \frac{M}{m}} + \frac{2M}{m} + \frac{64M \log(2/\beta)}{\varepsilon m}.$$

The lemmas follows by observing that if $h^* \in \underset{h \in \mathcal{H}}{\operatorname{argmin}} \, \widehat{R}_S^\rho(h)$, if and only if $h^* \in \underset{h \in \mathcal{H}}{\operatorname{argmin}} \, \widehat{R}_S^\rho(h) +$
$2\sqrt{\widehat{R}_S^\rho(h) \frac{M}{m}}$. $\qquad\qquad\qquad\qquad\qquad\qquad\qquad\qquad\qquad\qquad\qquad\qquad\qquad\qquad\qquad\quad\square$

## F  Confidence Margin Parameter Selection

The algorithms of Sections 3, 4, 5 and Appendix E can all be augmented to include the selection of the
confidence margin parameter $\rho$ by using an exponential mechanism. All of the proposed algorithms
in the previous sections output $w^{\mathsf{Priv}}$ such that

$$R_{\mathcal{D}}(w^{\mathsf{Priv}}) \leq F(\rho, g_\rho(S)),$$

where $g_\rho(S)$ is either the minimum $\rho$-margin loss or the minimum $\rho$-hinge loss. Furthermore, in all
our results, for any fixed $t$, $F(\rho, t)$ is a non-increasing function of $\rho$ and $g_\rho(S)$ is a non-decreasing
function of $\rho$ for any $S$. Suppose we have an algorithm $\mathcal{A}$ such that the above inequality holds.
We can then augment it with an exponential mechanism algorithm to select a near-optimal margin
$\rho$. Let $h_{\max}$ be an upper bound on $\max_{x \in \mathcal{X}} \max_{h \in \mathcal{H}} |h(x)|$. For example, $h_{\max} = \Lambda r$ for linear
classifiers. If $h_{\max} > \rho$, then the bound $F(\rho; g_\rho(S))$ becomes trivial (i.e., $\Omega(1)$). Similarly, the
bound typically becomes trivial when $\rho \lesssim \frac{h_{\max}}{\sqrt{m}}$. It is easy to see this property for linear classifiers,
for other models such as neural networks with label privacy it can be obtained by bounds on fat-
shattering dimension [BST99]. Hence, without loss of optimality, we will seek an approximation for
$\rho_{\mathsf{opt}}$ that minimizes $F(\rho; g_\rho(S))$ for $\rho \in \left[ \frac{h_{\max}}{\sqrt{m}}, h_{\max} \right]$. To do this, we define a finite grid over the
above interval: $\mathcal{V} \triangleq \left\{ \rho_j \triangleq 2^{-j} \, h_{\max} \; j \in [J] \right\}$, where $J = \frac{1}{2} \log(m)$. We use an instantiation of the
generalized exponential mechanism, with score function $-F(\rho; g_\rho(S))$, $\rho \in \mathcal{V}$ and privacy parameter
$\varepsilon$, to select $\rho^* \in \mathcal{V}$ that approximately minimizes $F(\rho; g_\rho(S))$ over $\rho \in \mathcal{V}$. We use the generalized
exponential mechanism as the sensitivity of $-F(\rho_j; g_{\rho_j}(S))$ depends on $\rho_j$. We then run $\mathcal{A}$ with
margin parameter $\rho = \rho^*$ to output the final parameter vector $w^{\mathsf{Priv}}$. For clarity, we include a formal
description of the full algorithm in Algorithm 6. We now state the guarantee of the augmented
algorithms.

**Lemma F.1.** *Let $\beta \in (0,1)$. Suppose $S \sim \mathcal{D}^m$ for some distribution $\mathcal{D}$ over $\mathcal{X} \times \mathcal{Y}$. Suppose $\mathcal{A}$ is $(\varepsilon, \delta)$ differentially private and its output satisfies $R_{\mathcal{D}}(w^{\mathsf{Priv}}) \leq F(\rho, g_\rho(S))$ with probability at least $1 - \beta$. Furthermore, for any $t$, let $F(\rho, t)$ be a non-increasing function of $\rho$ and $g_\rho(S)$ is a non-decreasing function of $\rho$ for any $S$. Then, Algorithm 6 is $(2\varepsilon, \delta)$-differentially private and with probability at least $1 - 2\beta$, the output $w^{\mathsf{Priv}}$ satisfies:*

$$R_{\mathcal{D}}(w^{\mathsf{Priv}}) \leq \min_{\rho \in \left[\frac{h_{\max}}{\sqrt{m}}, h_{\max}\right]} F(\rho/2, g_\rho(S)) + \frac{\Delta_{\rho/2}(F)}{\varepsilon} \cdot \log\left(\frac{\log(m)}{\beta}\right),$$

*where $\Delta_\rho(F)$ is a non-decreasing function of $\rho$ and is an upper bound on the sensitivity of $F$ given by $\Delta_\rho(F) = \max_{S, S': d(S, S')=1} |F(\rho, g_\rho(S)) - F(\rho, g_\rho(S'))|$ and $d(S, S')$ is the number of samples in which $S$ and $S'$ differ.*

*Proof.* The privacy guarantee follows from the basic composition property of differential privacy together with the fact that the generalized exponential mechanism invoked in step 2 is $\varepsilon$-differentially private and $\mathcal{A}$ is $(\varepsilon, \delta)$-differentially private.

We now turn to the proof of the error bound. Note that there exists $\hat{\rho} \in \mathcal{V}$ such that $\hat{\rho} \leq \rho_{\mathsf{opt}} < 2 \cdot \hat{\rho}$. By the properties of the generalized exponential mechanism [RS16, Theorem I.4] and the fact that the sensitivity of $F(\rho, g_\rho(S))$ is $\Delta_\rho(F)$, with probability at least $1 - \beta$ we have

$$
\begin{aligned}
F(\rho^*; g_{\rho^*}(S)) &\leq \min_{\rho \in \mathcal{V}} F(\rho; g_\rho(S)) + \frac{\Delta_\rho(F)}{\varepsilon} \cdot \log\left(\frac{\log(m)}{\beta}\right) \\
&\leq F(\hat{\rho}; g_{\hat{\rho}}(S)) + \frac{\Delta_{\hat{\rho}}(F)}{\varepsilon} \cdot \log\left(\frac{\log(m)}{\beta}\right) \\
&\leq F(\hat{\rho}; g_{\hat{\rho}}(S)) + \frac{\Delta_{\rho_{\mathsf{opt}}/2}(F)}{\varepsilon} \cdot \log\left(\frac{\log(m)}{\beta}\right) \\
&\leq F(\hat{\rho}; g_{\rho_{\mathsf{opt}}}(S)) + \frac{\Delta_{\rho_{\mathsf{opt}}/2}(F)}{\varepsilon} \cdot \log\left(\frac{\log(m)}{\beta}\right) \\
&\leq F(\rho_{\mathsf{opt}}/2; g_{\rho_{\mathsf{opt}}}(S)) + \frac{\Delta_{\rho_{\mathsf{opt}}/2}(F)}{\varepsilon} \cdot \log\left(\frac{\log(m)}{\beta}\right), \quad (30)
\end{aligned}
$$

where the last two inequalities follow from the fact that for any $t$, let $F(\rho, t)$ be a non-increasing function of $\rho$ and $g_\rho(S)$ is a non-decreasing function of $\rho$ for any $S$. By the assumption on $\mathcal{A}$, with probability $1 - \beta$,

$$R_{\mathcal{D}}(w^{\mathsf{Priv}}) \leq F(\rho^*; g_{\rho^*}(S)). \quad (31)$$

Combining (30) and (31) yields the lemma. The error probability follows by the union bound. $\square$

The above lemma can be combined with any of the algorithms of Section 3, 4 and Appendix E. We instantiate it for $\mathcal{A}_{\mathsf{EffPrivMrg}}$ in the following corollary. Below, we compute sensitivity for the bounds on other algorithms, which can be used to get similar guarantees.

**Corollary F.2.** *Let $\beta \in (0,1)$ and $m \in \mathbb{N}$. Suppose $S \sim \mathcal{D}^m$ for some distribution $\mathcal{D}$ over $\mathcal{X} \times \mathcal{Y}$. Recall that by Theorem 3.2, the output of Algorithm 2 (denoted by $w'$) with probability at least $1 - \beta/2$ satisfies, $R_{\mathcal{D}}(w') \leq F(\rho, g_\rho(S))$, where $g_\rho(S) = \min_{w \in \mathbb{B}^d(\Lambda)} \widehat{L}_S^\rho(w)$ and*

$$F(\rho', t) = t + O\left(\sqrt{\frac{\log(1/\beta)}{m}} + \frac{\Lambda r}{\rho'}\left(\frac{1}{\sqrt{m}} + \frac{\sqrt{\log(\frac{m}{\beta})\log(\frac{1}{\beta})}\log^{\frac{3}{4}}(\frac{1}{\delta})}{\sqrt{\varepsilon m}}\right)\right).$$

*Let $w^{\mathsf{Priv}}$ be the output of Algorithm 6 with inputs $S$, privacy parameters $\varepsilon/2, \delta$, bound $F(\rho, g_\rho(S))$, algorithm $\mathcal{A}_{\mathsf{EffPrivMrg}}$, confidence parameter $\beta/2$, and $h_{\max} = \Lambda r$. Then $w^{\mathsf{Priv}}$ is $(\varepsilon, \delta)$ differentially private. Furthermore, with probability at least $1 - \beta$,*

$$R_{\mathcal{D}}(w^{\mathsf{Priv}}) \leq \min_{\rho \in \left[\frac{\Lambda r}{\sqrt{m}}, \Lambda r\right]} F(\rho/2, g_\rho(S)) + O\left(\frac{\Lambda r}{m\rho\varepsilon} \cdot \log\left(\frac{\log(m)}{\beta}\right)\right).$$

**Lemma F.3.** *Fix $\rho > 0$. Let the functions $F_1$, $F_2$, $F_3$, $F_4$ and $F_5$ are defined as follows:*

$$F_1(\rho', g_\rho(S)) = \min_{w \in \mathbb{B}^d(\Lambda)} \widehat{R}_S^\rho(w) + O\left( \sqrt{\widehat{R}_S^\rho(w)\left( \frac{\Lambda^2 r^2 \log^2\left(\frac{m}{\beta}\right)}{m(\rho')^2} + \frac{\log\left(\frac{1}{\beta}\right)}{m} \right)} + \Gamma \right),$$

$$F_2(\rho', g_\rho(S)) = \min_{w \in \mathbb{B}^d(\Lambda)} \widehat{L}_S^\rho(w) + O\left( \sqrt{\frac{\log(1/\beta)}{m}} + \frac{\Lambda r}{\rho'}\left( \frac{1}{\sqrt{m}} + \frac{\sqrt{\log(\frac{m}{\beta})\log(\frac{1}{\beta})}\log^{\frac{3}{4}}(\frac{1}{\delta})}{\sqrt{\varepsilon m}} \right) \right),$$

$$F_3(\rho', g_\rho(S)) = \min_{h \in \mathcal{H}_\Lambda} \widehat{L}_S^\rho(h) + O\left( \sqrt{\frac{\log(1/\beta)}{m}} + \frac{\Lambda r}{\rho'}\left( \frac{1}{\sqrt{m}} + \frac{\sqrt{\log(\frac{m}{\beta})\log(\frac{1}{\beta})}\log^{\frac{3}{4}}(\frac{1}{\delta})}{\sqrt{\varepsilon m}} \right) \right),$$

$$F_4(\rho', g_\rho(S)) = \min_{h \in \mathcal{H}} \widehat{R}_S^\rho(h) + 2\sqrt{\min_{h \in \mathcal{H}} \widehat{R}_S^\rho(h)}\sqrt{\frac{M}{m}} + \frac{2M}{m} + \frac{64M \log\left(\frac{2}{\beta}\right)}{\varepsilon m},$$

$$F_5(\rho', g_\rho(S)) = \min_{h \in \mathcal{H}_{\mathrm{NN}\Lambda}} \widehat{R}_S^\rho(h) + O\left( \frac{r(2\eta\Lambda)^L \sqrt{N\theta}}{\rho'\sqrt{m}} + \frac{r^2(2\eta\Lambda)^{2L} N\theta}{(\rho')^2 \varepsilon m} \right),$$

*where $M$ is defined in Theorem E.1 and $\Gamma$ is defined in Theorem 3.1. Then*

$$\Delta_\rho(F_1) = O\left( \frac{1}{m} + \frac{1}{m}\sqrt{\frac{\Lambda^2 r^2 \log^2\left(\frac{m}{\beta}\right)}{\rho^2} + \log\left(\frac{1}{\beta}\right)} \right).$$

$$\Delta_\rho(F_2) = O\left( \frac{\Lambda r}{m\rho} \right).$$

$$\Delta_\rho(F_3) = O\left( \frac{\Lambda r}{m\rho} \right).$$

$$\Delta_\rho(F_4) = O\left( \frac{1 + \sqrt{M}}{m} \right).$$

$$\Delta_\rho(F_5) = O\left( \frac{1}{m} \right).$$

*Proof.* We provide the proof for the bound on $\Delta_\rho(F_2)$. The proof for other quantities is similar and omitted. Let $S'$ and $S''$ be two samples that differ in at most one sample. Without loss of generality, let $F_1(\rho, g_\rho(S')) \geq F_2(\rho, g_\rho(S''))$. Let

$$w' \in \operatorname*{argmin}_{w \in \mathbb{B}^d(\Lambda)} \widehat{L}_{S'}^\rho(w) + O\left( \sqrt{\frac{\log(1/\beta)}{m}} + \frac{\Lambda r}{\rho}\left( \frac{1}{\sqrt{m}} + \frac{\sqrt{\log(\frac{m}{\beta})\log(\frac{1}{\beta})}\log^{\frac{3}{4}}(\frac{1}{\delta})}{\sqrt{\varepsilon m}} \right) \right)$$

and

$$w'' \in \operatorname*{argmin}_{w \in \mathbb{B}^d(\Lambda)} \widehat{L}_{S''}^\rho(w) + O\left( \sqrt{\frac{\log(1/\beta)}{m}} + \frac{\Lambda r}{\rho}\left( \frac{1}{\sqrt{m}} + \frac{\sqrt{\log(\frac{m}{\beta})\log(\frac{1}{\beta})}\log^{\frac{3}{4}}(\frac{1}{\delta})}{\sqrt{\varepsilon m}} \right) \right).$$

Then

$$\begin{aligned}
F_1(\rho, g_\rho(S')) - F_2(\rho, g_\rho(S'')) &= \widehat{L}_{S'}^\rho(w') - \widehat{L}_{S''}^\rho(w'') \\
&\leq \widehat{L}_{S'}^\rho(w'') - \widehat{L}_{S''}^\rho(w'') \\
&\leq \frac{2}{m\rho} \max_{w \in \mathbb{B}^d(\Lambda), x \in \mathbb{B}^d(r)} |w \cdot x| \\
&\leq \frac{2}{m\rho}\Lambda r.
\end{aligned}$$

$\square$

# G  Example of high error for exponential mechanism

**Lemma G.1.** *Let $d \geq c$ for some constant $c$ and $\rho \in [0,1]$. There exists a distribution $\mathcal{D}$ over $\mathbb{B}^d$ and a subset $\mathcal{H} \in \mathcal{H}_{\mathsf{Lin}}$ such that the following hold:*

- *__Realizable setting__: There exists a $h^*$ in $\mathcal{H}$ such that $R_{\mathcal{D}}(h^*) = 0$.*

- *__Only one good hypothesis__: For any $h$ in $\mathcal{H} \smallsetminus \{h^*\}$, $R_{\mathcal{D}}(h) = 1$.*

- *__A good cover__: For any two $h_w, h_{w'} \in \mathcal{H}$, their corresponding weights satisfy $|\langle w, w' \rangle| \geq 1/8$.*

- *__Exponential mechanism incurs high error__: Given $m < c' \cdot d/\varepsilon$ samples $\mathcal{D}$, with probability at least $9/10$, the exponential mechanism on $-\widehat{R}_S^\rho(h)$ will select a $h$ such that $R_{\mathcal{D}}(h) = 1$.*

*Proof.* Let $\mathcal{D}$ be defined as follows. Let $\mathcal{D}(x)$ be a uniform distribution over $\{-1/\sqrt{d}, 1/\sqrt{d}\}^d$ and $y = 1$ if $x_1 > 0$, $0$ otherwise. The optimal hypothesis $h^*(x) = 1_{x_1>0}$ and satisfies $R_{\mathcal{D}}(h^*) = 0$. Let $\mathcal{H} = \{h^*\} \cup \{h_w : w \in \mathcal{W}\}$, where $\mathcal{W}$ is the largest set such that for all $w \in \mathcal{W}$, $w_1 = -1/\sqrt{d}$ and for any two $w, w' \in \mathcal{W}$, $|\langle w, w' \rangle| \geq 1/8$. By the Gilbert-Varshamov bound, the size of such a set is at least $2^{c \cdot d}$ for some constant $c$. Note that for any $\mathcal{H} \smallsetminus \{h^*\}$, $R_{\mathcal{D}}(h) = 1$.

Now suppose we use the exponential mechanism with score $-\widehat{R}_S^\rho(h)$. The probability of selecting the correct hypothesis is at most

$$\frac{1}{\sum_{h \in \{h_w : w \in \mathcal{W}\}} \exp(-\widehat{R}_S^\rho(h)\varepsilon/2m)} \leq \frac{1}{2^{c \cdot d} e^{-\varepsilon m/2}} = e^{\varepsilon m/2 - c'd}.$$

Hence if $m < c'/d\varepsilon$, then the probability of choosing $h^*$ is at most $e^{-c'd/2} \leq 1/10$ for $d \geq \frac{2}{c'} + 3$. $\qquad\square$

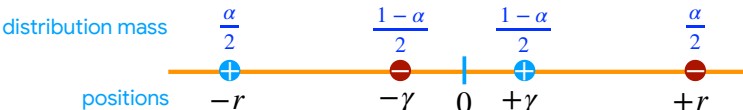

**Figure 3:** Simple example in dimension one for which the minimizer of the expected hinge loss $\mathbb{E}[\ell^{\text{hinge}}(w)]$ is $w^* = \frac{1}{\gamma}$ and thus $\|w^*\| = \frac{1}{\gamma} \gg 1$ for $\gamma \ll 1$.

## H  Example of a large norm hinge-loss minimizer

Fix $\alpha \in [0, 1]$ and $\gamma \in (0, r)$. Consider the distribution $\mathcal{D}$ on the real line (dimension one) defined as follows: there is a probability mass of $\frac{\alpha}{2}$ at coordinate $(+r, -1)$, a probability mass of $\frac{\alpha}{2}$ at $(-r, +1)$, a probability mass of $\frac{1-\alpha}{2}$ at $(+\gamma, +1)$, and a probability mass of $\frac{1-\alpha}{2}$ at $(-\gamma, -1)$. Figure 3 illustrates this distribution. We first examine the expected hinge loss $\ell^{\text{hinge}}(w)$ of an arbitrary linear classifier $w \in \mathbb{R}$ in dimension one:

$$\ell^{\text{hinge}}(w) = \frac{\alpha}{2}[\max\{0, 1 + wr\} + \max\{0, 1 + wr\}] + \frac{1 - \alpha}{2}[\max\{0, 1 - w\gamma\} + \max\{0, 1 - w\gamma\}]$$
$$= \alpha \max\{0, 1 + wr\} + (1 - \alpha)\max\{0, 1 - w\gamma\}.$$

Thus, distinguishing cases based on the value of scalar $w$, we can write:

$$\ell^{\text{hinge}}(w) = \begin{cases} \alpha(1 + wr) + (1 - \alpha)(1 - w\gamma) & \text{if } w \in \left[0, \frac{1}{\gamma}\right] \\ \alpha(1 + wr) & \text{if } w \geq \frac{1}{\gamma} \\ \alpha(1 + wr) + (1 - \alpha)(1 - w\gamma) & \text{if } w \in \left[-\frac{1}{r}, 0\right] \\ (1 - \alpha)(1 - w\gamma) & \text{if } w \leq -\frac{1}{r}. \end{cases}$$
$$= \begin{cases} w(\alpha r - (1 - \alpha)\gamma) + 1 & \text{if } w \in \left[0, \frac{1}{\gamma}\right] \\ \alpha(1 + wr) & \text{if } w \geq \frac{1}{\gamma} \\ w(\alpha r - (1 - \alpha)\gamma) + 1 & \text{if } w \in \left[-\frac{1}{r}, 0\right] \\ (1 - \alpha)(1 - w\gamma) & \text{if } w \leq -\frac{1}{r}. \end{cases}$$

To simplify the discussion, we will set $r = 1$ and $\alpha = \frac{\gamma}{2}$, with $\gamma \ll 1$. This, implies $(\alpha r - (1 - \alpha)\gamma) = \alpha(-1 + 2\alpha) < 0$. As a result of this negative sign, the best solution for the first two cases above is $w = \frac{1}{\gamma}$, $w = 0$ in the third case, and $w = -\frac{1}{r}$ in the last case. The loss achieved in the two latter cases is 1 and $(1 - \alpha)(1 + \frac{\gamma}{r})$, both larger than the loss $\alpha(1 + \frac{r}{\gamma})$ obtained in the first two cases. In view of that, the overall minimizer of $\ell^{\text{hinge}}(w)$ is given by $w^* = \frac{1}{\gamma}$, with $\ell^{\text{hinge}}(w^*) = \frac{1}{2}(1 + \gamma)$. Note that the zero-one loss of $w^*$ is $\ell(w^*) = \alpha = \frac{\gamma}{2}$.

Thus, for this example, the norm of the hinge-loss minimizer is arbitrary large: $\|w^*\| = \frac{1}{\gamma} \gg 1$. In particular, for a sample size $m$, we could choose $\gamma < \frac{1}{m}$, leading to $\|w^*\| > m$. Note that, here, any other positive classifier, $w > 0$, achieves the same zero-one loss as $w^*$. For example, $w = 1$ achieves the same performance as $w^*$ with a more favorable norm.

Our analysis was presented for the population hinge loss but a similar result holds for the empirical hinge loss.