# OpenReview forum: "Differentially Private Learning with Margin Guarantees"
_NeurIPS.cc/2022/Conference — NeurIPS 2022 Accept_

### Official Review · Reviewer_NXiP · 2022-06-20

**Rating:** 5
**Confidence:** 3
**Soundness:** 3 good
**Presentation:** 3 good
**Contribution:** 3 good

**Summary:**

This paper proposes a series of DP algorithms for learning linear classifiers, kernel classifiers, and neural-network classifiers with dimension-independent and confidence-margin guarantees. It leverages JL-transform and a faster DP-ERM algorithm to design the faster DP linear classifier with margin guarantees. It combines kernel approximation with the use of the JL-transform to design the DP kernel classifiers with margin guarantees. And in the end, the authors propose a DP neural network learning algorithm for a family of feed-forward neural networks for which they prove a confidence-margin bound that is independent of the input dimension via embedding-based ``network compression'' technique. This enables a broader analysis of DP neural network in the future.

**Questions:**

1. Is it possible to apply advanced composition during the privacy leakage analysis?

**Limitations:**

Yes. The authors have addressed  potential negative societal impact.

**Strengths And Weaknesses:**

Strengths:
1. This paper introduces its DP algorithms step by step from the DP linear classifer, to the DP kernel classifier and finally the DP neural nework with a very large input dimension and margin guarantees, which makes it easier to understand.
2. Using fast JL-transform to reduce the high dimensionality to improve the time complexity is a nice optimization.
3. Using the embedding-based network compreesion technique to improve the DP neural network learning algorithm.

Weakness:
1. There is no experiments to enhance the theoretical conclusions. It will be better even if the authors can run toy experiments on some small datasets.
2.  It is possible to improve the privacy leakage via advanced composition.

---

> ### Author Response · Authors · 2022-08-02
> **Response to Reviewer NXiP**
>
> Thank you for the review. Please, see our response to your comments and questions below.
>
> -  “There is no experiments to enhance the theoretical conclusions. It will be better even if the authors can run toy experiments on some small datasets.”
>
>
> Our contributions are theoretical and we believe that the fundamental nature of the problem we address makes this work interesting to the community. We will report empirical results in future work.
>
>
> -  “Is it possible to apply advanced composition during the privacy leakage analysis?”
>
>
> First, regarding Algorithms 2 and 3 (in Sections 3 and 4, respectively), note that the privacy guarantee follows from the privacy guarantee of the DP-ERM subroutine (Algorithm 4 in Appendix B.2 (invoked in step 4 in Algorithm 2)). Algorithm 4 consists of $M$ rounds, where $M=\log(1/\beta)$ and $\beta$ is the confidence parameter. As discussed in Appendix B.2, the privacy analysis of Algorithm 4 entails three main ingredients: i) we first show that each round is $(\frac{\varepsilon}{2M}, \frac{\delta}{M})$-DP by combining the privacy guarantee of Algorithm $\mathcal{A}_{\mathsf{GLL}}$ from [BGM21] (invoked in Step 6) together with the group privacy property of DP, ii) we then apply basic composition over the $M$ rounds, and finally iii) compose the resulting privacy guarantee with the privacy guarantee of the exponential mechanism invoked in step 7. Note that since $M$ is logarithmic in $1/\beta$, applying advanced composition over the $M$ rounds does not buy us any meaningful improvement in the final bound. In particular, if we apply advanced composition, then we have $\epsilon \approx \hat{\epsilon}~\sqrt{M \log(1/\delta)}$, and hence, the $\log(1/\beta)$ term in the final bound becomes $\sqrt{\log(1/\beta) \log(1/\delta)}$. Note also that the latter bound is worse than the former (our bound) when $\delta < \beta$ (which is a common setting since $\delta$ can be viewed as the probability of a catastrophic privacy breach).
>
> The remaining algorithms in the paper (the pure DP algorithms) are based on private selection from a finite cover of the hypothesis class via the exponential mechanism. Applying advanced composition does not make sense in this case since there are no rounds to compose over.

---

### Official Review · Reviewer_kUdN · 2022-06-25

**Rating:** 6
**Confidence:** 4
**Soundness:** 3 good
**Presentation:** 3 good
**Contribution:** 3 good

**Summary:**

This paper develops several differentially private algorithms with dimension-independent utility guarantees. The results include a pure DP-algorithm and an efficient private algorithm for linear classification. The paper further extends these results to kernel-based hypotheses with shift-invariant kernels and feed-forward neural networks. For all the algorithms, the paper develops margin guarantees independent of the dimension.

**Questions:**

Question
- In the introduction, the authors say that learning guarantee necessarily admits a dependency on $d$ for constrained setting. However, it seems that the paper considers a constrained problem setting: Algorithm 2 involves a minimization over $B^k(2\Lambda)$, the linear predictors are also constrained as $w\in B^d(\Lambda)$. Therefore, it is a bit confusing to me whether the dimension-independent bounds contradict with existing results.
- The paper uses the technique of margin and random projection. It is not clear to me which one is essential to develop dimension-independent guarantee. It would be helpful if the authors could comment on how the margin technique is useful in improving the DP guarantee.

Typos:

Section 4: $(\tilde{\Psi}(x),y))$

Section 4: can designed

Section 4: builds on on

Lemma A.2: the meaning of fat is not rigorous

**Limitations:**

Yes.

**Strengths And Weaknesses:**

Strength:

The existing DP algorithms suffer from a bound with a dependency on either the dimension or the norm of the optimal model, which would not be effective if the dimension or the norm of the optimal model is large. This paper shows that these dependency can be removed and develops several algorithms based on margin guarantees. Therefore, the results are interesting to the differential privacy community.

Weakness:
- The bound in Theorem 5.1 suffers from an exponential dependency on the depth. Therefore, the result is not efficient if $L$ is moderately large.
- For the pure DP algorithm, the bounds have a quadratic dependency on $1/\rho$. The bounds would not be effective if $\rho$ is small.
- The paper presents a series of results. While these results are comprehensive, they may make the paper not focused.

---

> ### Author Response · Authors · 2022-08-02
> **Response to Reviewer kUdN**
>
> Thank you for the careful review and for your comments. Please, see our response to your comments and questions below.
>
> -  “The bound in Theorem 5.1 suffers from an exponential dependency on the depth.”
>
> Please note here that standard learning bounds based on uniform convergence also include a term of the form $\Lambda^L$. We also note that recent margin bounds (in the non-private setting) such as those in [BFT17] involve a similar dependency (particularly, the product of the norms of the weight matrices of all the layers).
>
> [BFT17]: Peter Bartlett, Dylan Foster, Matus Telgarsky. Spectrally-normalized margin bounds for neural networks. NeurIPS 2017.
>
>
>
>
>
> -  “For the pure DP algorithm, the bounds have a quadratic dependency…”
>
> The mentioned dependency appears in the error term due to privacy, which scales as $1/(\rho^2 m)$. Please, note that this term is essentially the square of the first term $1/(\rho \sqrt{m})$, which is the standard non-private bound. That is, the error term due to privacy is essentially dominated by the standard non-private error (assuming  $\epsilon = \Omega(1)$).
>
>
> -  “The paper presents a series of results. While these results are comprehensive, they may make the paper not focused.”
>
> Deriving dimension-independent guarantees in privacy-preserving learning is a fundamental problem and our results and solutions are new. Thus, we aimed to provide a more comprehensive set of results to show how different aspects can be tackled. Nevertheless, we are happy to follow any suggestion from the reviewer to improve our presentation.
>
>
> -  “… it seems that the paper considers a constrained problem setting... Therefore, it is a bit confusing to me whether the dimension-independent bounds contradict with existing results.”
>
> Despite having a bound $\Lambda$ on the norm given as an input parameter to the algorithm, we do not subject our algorithm’s final output to that constraint. In particular, note that in the final step of our algorithms, we return a predictor in the original $d$-dimensional space by applying the transpose of the original projection matrix $\Phi^{\top} \tilde{w}$. The final output $\Phi^{\top} \tilde{w}$ has expected norm $\sqrt{d}\rho/r$ and may not lie in $B^d(\Lambda)$. Nevertheless, the margin guarantee obtained only depends on $\Lambda/\rho$, not on the norm of the output. This enables us to circumvent the dependency on the dimension.
>
>
> -  “The paper uses the technique of margin and random projection. It is not clear to me which one is essential to develop dimension-independent guarantee…”
>
> To answer this question precisely, let us emphasize first that, even in the non-private setting, margin bounds are critical in classification problems to derive dimension-independent (and VC-dimension-independent) guarantees. The challenge here has been to derive private algorithms benefiting from such guarantees. Random projection is one of the essential techniques used in our algorithms and analyses to achieve that goal. As an alternative to random projection, as indicated in section 4, we can use sketching techniques for that purpose. As in the non-private setting, margin guarantees provide us with a trade-off between the empirical margin loss and complexity, depending on the confidence parameter $\rho$. In summary, both margin guarantees and random projection play a critical role here, indeed.
>
> -  We also thank the reviewer for pointing out these typos. We will fix them in the final version of the paper.

---

### Official Review · Reviewer_kj5q · 2022-07-02

**Rating:** 7
**Confidence:** 2
**Soundness:** 3 good
**Presentation:** 3 good
**Contribution:** 3 good

**Summary:**

This paper introduces several new differentially private learning algorithms with performance guarantees that are independent from the data input dimension and instead given in terms of a confidence margin parameter.
Algorithms are presented for linear classification, kernel classifiers, and neural networks. One of the key tools of the analysis is to use the JL transform to map the hypotheses to a lower-dimensional space, perform learning there, and then to “invert” the transform.

**Questions:**

What is $r$ in Theorem 5.1? It appears undefined in the main text.
Can the algorithm for learning the NN be efficiently implemented? What is the significance of the bounded norm assumption?

Minor comments: The sentence in lines 102-104 should be fixed
111 and 112: yields/makes: remove s
128: “gave construction”
138 and 139: the term “$\rho$-hinge loss” is overloaded
Alg. 1 line 4: “an”
186: “empirical $\rho$-margin” should be “empirical $\rho$-margin loss”?

**Limitations:**

The impact of the assumptions that are made is not extensively discussed in the paper.

**Strengths And Weaknesses:**

This paper provides interesting results regarding a relevant topic. While many of the main ideas appear to be similar to prior work, the analysis requires non-standard technical extensions, and the resulting bounds are clear improvements.
The models that are studied cover a wide variety of use cases, and both inefficient algorithms, but which are stronger in terms of privacy guarantees, as well as computationally efficient algorithms, at the cost of weaker privacy guarantees, are considered.

In terms of weaknesses, the practicality of the results and limitations could be discussed to a further extent. The bound in Theorem 5.1 could be further discussed to elucidate the role of the parameters—it appears that $r$ is not even defined here.

---

> ### Author Response · Authors · 2022-08-02
> **Response to Reviewer kj5q**
>
> We thank the reviewer for the valuable feedback. Below, we respond to the questions raised by the reviewer.
>
> -  "What is $r$ in Theorem 5.1?"
>
> The parameter $r$ is defined in line 311 in the beginning of Section 5. Namely, $r$ is the norm bound on the feature vectors. This is also consistent with the notation we use for this parameter throughout the paper. Please, note that the presence of that parameter is also standard in non-private learning bounds.
>
>
> -  "Can the algorithm for learning the NN be efficiently implemented?"
>
> This is a good question. The fundamental limitation here stems from the computational hardness of the non-convex optimization problem underlying NN learning (even non-privately). This is due to the non-convexity of the empirical objective in the case of NN learning (even if we replace the zero-one loss with a more well-behaved function such as the hinge loss). There are alternatives that seek either a first-order or second-order stationary point of the non-convex objective; however, since our main focus is to obtain an information-theoretic generalization bound for private NN learning, we did not consider such alternatives here.
>
>
> -  "What is the significance of the bounded norm assumption?"
>
> We note here that the bound $\Lambda$ on the norm is a measurable and, more importantly, a tunable quantity, for which one can select an optimal setting using model selection techniques. Existing non-private margin bounds (whether for linear and kernel-based classifiers [MRT18] or NN learning [BFT17]) fundamentally entail a similar dependency. For our margin bound for NN learning, we can treat $\Lambda^L / \rho$ as a tunable parameter as discussed in the paragraph given by lines 198-202.
>
> [MRT18]: Mehryar Mohri, Afshin Rostamizadeh, and Ameet Talwalkar. Foundations of machine learning. MIT press, 2018.
>
> [BFT17]: Peter Bartlett, Dylan Foster, Matus Telgarsky. Spectrally-normalized margin bounds for neural networks. NeurIPS 2017.
>
>
> -  We also thank the reviewer for pointing out these typos. We will fix them in the final version of the paper.

---

### Official Review · Reviewer_6Kj9 · 2022-07-11

**Rating:** 7
**Confidence:** 2
**Soundness:** 4 excellent
**Presentation:** 4 excellent
**Contribution:** 4 excellent

**Summary:**

The paper presents differentially private algorithms with dimension-independent margin guarantees for various classification problems such as linear classification, kernel-based classification, and feed-forward neural network classification. The key to the algorithm for linear classification is using the FAST JL transform for feature space reduction and then using the DP-ERM algorithm from  [BGM21] for solving the convex optimization problem (along with standard boosting techniques). The key to the algorithm for private kernel-based classification is to first approximate the feature map by a finite-dimensional map determined via Random Fourier Features. Finally, for  Feed Forward Neural Networks classification, the authors rely once again on the JL transform to reduce the dimensionality before each layer and apply the exponential mechanism to achieve pure differential privacy.

Overall, this is a compressive and systematic study on differentially private classification and I recommend it be expected.

**Questions:**

None.

**Limitations:**

Yes.

**Strengths And Weaknesses:**

The paper provides a bunch of elegant algorithms for differentially private classification. While the technical components used in the algorithms are well known, the literature review, as well as results, are comprehensive.

---

> ### Author Response · Authors · 2022-08-02
> **Response to Reviewer 6Kj9**
>
> We thank the reviewer for their comments and appreciation of our work.

---

### Meta-Review · Area_Chair_qpNg · 2022-08-26

**Recommendation:** Accept
**Confidence:** Certain

**Metareview:**

After the internal discussion, all reviewers agreed that the paper should be accepted. Please take into account the reviewers' comments while preparing the camera-ready version of the paper.

**Award:**

No

---

### Decision · Program_Chairs · 2022-09-14

Accept